## TOOLS

# DeepContact: High-throughput quantification of membrane contact sites based on electron microscopy imaging

Liqing Liu[1,5]*, Shuxin Yang[2,4]*, Yang Liu[2,4], Xixia Li[5], Junjie Hu[1,6], Li Xiao[2,3,4], and Tao Xu[1,6,7]

**Membrane contact site (MCS)-mediated organelle interactions play essential roles in the cell. Quantitative analysis of MCSs reveals vital clues for cellular responses under various physiological and pathological conditions. However, an efficient tool is lacking. Here, we developed DeepContact, a deep-learning protocol for optimizing organelle segmentation and contact analysis based on label-free EM. DeepContact presents high efficiency and flexibility in interactive visualizations, accommodating new morphologies of organelles and recognizing contacts in versatile width ranges, which enables statistical analysis of various types of MCSs in multiple systems. DeepContact profiled previously unidentified coordinative rearrangements of MCS types in cultured cells with combined nutritional conditions. DeepContact also unveiled a subtle wave of ER–mitochondrial entanglement in Sertoli cells during the seminiferous epithelial cycle, indicating its potential in bridging MCS dynamics to physiological and pathological processes.**

## Introduction

Eukaryotic cells are compartmentalized via intracellular membranes into organelles with distinct physiological functions. Precise coordination among organelles is essential to enable the operation of the cell as a functional unit. Direct physical interactions between organelles by molecular tethering, namely membrane contact sites (MCSs; Scorrano et al., 2019), have been found to regulate the cellular homeostasis of lipids (Jeong et al., 2017; Lahiri et al., 2015; Tong et al., 2018), calcium ions (Burgoyne et al., 2015; Krols et al., 2016), and reactive oxygen species (Eisner et al., 2013) and transmit signals (Toulmay and Prinz, 2011) and forces (Rowland et al., 2014) for organelle dynamics (Namba, 2019; Prinz et al., 2019). There is an increasing demand to elucidate the precise roles of MCSs in various physiological and pathological conditions (Bohnert, 2020; Helle et al., 2013; Horvath et al., 2015; Phillips and Voeltz, 2016; Prinz, 2014).

EM acquires ultraresolution landscape images of subcellular contents, presenting the most direct morphological information on organelles, as well as their interactions, in nanoscale detail. Advances in EM have provided critical insight into the features of MCSs (Fernández-Busnadiego et al., 2015; Meschede et al., 2020). For example, proteinaceous tethers between the ER and

mitochondria, ranging from 10 to 25 nm at the smooth ER and from 50 to 80 nm at the rough ER, have been defined by EM (Csordás et al., 2006; Giacomello and Pellegrini, 2016). Quantitative analysis of EM images has provided vital clues about the involvement of MCSs in specific cellular events (Zhao et al., 2017). However, manual annotation–based organelle segmentation is labor-intensive and biased in defining organelle parameters, and it relies on sufficient imaging resolution and accurate determination of the organelle boundary. In addition, precise MCS profiling demands a high-throughput system to account for variations between samples (Bag et al., 2020; Scorrano et al., 2019).

Deep learning methods have been developed to recognize organelles in EM images, mostly one at a time (Haberl et al., 2018; Xiao et al., 2018; Zhang et al., 2019). Advancement of EM imaging techniques and computing resources allow deep learning–based 3D analysis of MCSs based on organelle segmentation in high-resolution volumetric data, which demands days of machine time for EM imaging and high-performance computing resources for the analysis of a single cell (Heinrich et al., 2021; Liu et al., 2020). However, cross-sample comparisons of biological samples can easily be complicated by the

[1]National Laboratory of Biomacromolecules, Institute of Biophysics, Chinese Academy of Sciences, Beijing, China; [2]Key Laboratory of Intelligent Information Processing, Institute of Computing Technology, Chinese Academy of Sciences, Beijing, China; [3]Ningbo HuaMei Hospital, University of Chinese Academy of Sciences, Ningbo, China; [4]School of Computer and Control Engineering, University of Chinese Academy of Sciences, Beijing, China; [5]Center for Biological Imaging, Institute of Biophysics, Chinese Academy of Sciences, Beijing, China; [6]College of Life Science, University of Chinese Academy of Sciences, Beijing, China; [7]School of Biomedical Engineering, Guangzhou Medical University, Guangzhou, Guangdong, China.

*L. Liu and S. Yang contributed equally to this paper. Correspondence to Tao Xu: xutao@ibp.ac.cn; Li Xiao: xiaoli@ict.ac.cn; Junjie Hu: huj@ibp.ac.cn.



heterogeneity of the cells (Yang et al., 2018), requiring a large sample size for deep learning–based statistical analysis, which hardly adapts to the time and resource demands of 3D analysis at the current stage.

We designed DeepContact, a deep learning protocol for MCS analysis based on accurate segmentation of individual organelles by model optimization, with high efficiency and flexibility to accommodate various types and morphologies of organelles in various EM samples. The system automates the procedure of processing the 2D EM slices, segmenting and visualizing MCSs in versatile width ranges, and quantifying normalized MCS parameters in large sample sizes, enabling cross-sample contact analysis in various biological processes. The system was verified in both cultured cells and specific cell types within tissue samples, revealing unprecedented throughput and details of MCS profiling in different physiological settings. With a computationally accessible setup, our pioneering work provides an efficient and comprehensive approach to high-throughput statistical analyses of MCSs in vitro and in vivo.

## Results

### Design of DeepContact

To automatically segment organelles and compute their MCSs by defining the distance between the cytosolic faces of the organelles (contact width), we established a deep learning–based method using EM images. The procedure was demonstrated initially by analyzing the ER-mitochondria (ER-Mito) contact (Fig. 1).

EM images can be different sizes and resolutions. To standardize the analysis procedure and reduce the computational cost of high-resolution images, we preprocessed images by extracting the region of interest (RoI) and resized them into standard-sized patches of 1,024 × 1,024 pixels with a resolution of 10 nm. Subsequently, image intensity was normalized. In the training stage, we used a sliding window protocol to extract 1,024 × 1,024-pixel patches with a 512-pixel step size, and affine transform and Gaussian blur were applied to each patch to generate the training samples. These samples were then used to train the DeepContact models. Segmenting the ER at image level may help model the integrative patterns of the ER and exclude other cisterna-like organelle networks from the model, particularly the Golgi apparatus. Therefore, we tested several semantic segmentation models, including U-Net (Ronneberger et al., 2015), LinkNet (Chaurasia and Culurciello, 2017), FPN (Lin et al., 2016), and PSPnet (Zhao et al., 2016), and their performances are summarized in Table S1. DeepContact adopts U-Net (Ronneberger et al., 2015) as the backbone for ER segmentation because it achieves the best performance on both Dice and mean intersection over union (mIoU) metrics. Meanwhile, we adopted instance segmentation model Mask R-CNN (He et al., 2017) for mitochondrial segmentation, as mitochondria are composed of physiologically and functionally individualized entities, and DeepContact needs to segment each mitochondrion for contact analysis. To avoid false-positive predictions caused by the similarity between targeted mitochondria and other organelles, a top likelihood loss (Xiao et al., 2019) was implanted in

the Mask R-CNN to improve model performance by sampling the most suspected target regions during training. A similarity loss was then added for improved feature identification of the positive and negative proposals (Fig. S1 a).

We randomly cropped five patches on each image during inference, segmented the organelles, and determined the boundaries of the targeting organelles on each patch. We programmed an automatic recording and calculation of organelle information, including numbers, perimeters, areas, and, most importantly, the MCS profile. In addition, the width of each contact point, based on pixel units, was measured and recorded. Quantification of the ER-Mito contact was usually normalized by mitochondrial parameters (Yang et al., 2018; Zhao et al., 2017). The MCS profile was defined as the corresponding pixels lining the outer membrane of the mitochondria in the current study. Therefore, the ER-Mito contact ratio is computed as Mito_boundary contact divided by Mito_boundary. Mito_boundary was computed by summing all of the mitochondria boundaries among the patches from a single EM image, and similarly, Mito_boundary contact was the sum of all contacting mitochondria boundaries among the patches from a single EM image. The same principle was used in the lipid droplet (LD)-Mito contact calculation in subsequent applications.

To verify the accuracy of DeepContact in segmenting organelles, as a gold standard, a senior expert elaborately annotated six images of U-2 OS cells, which were taken from six different cells in one section of a new EM sample block that had never been involved in the training processes. We then randomly cropped five patches from each image, 30 patches in total, to make up the testing set. We defined the metric match ratio, which is the total number of correctly predicted pixels divided by the total number of annotated pixels, to quantify the prediction accuracy. The mean match ratio of mitochondria, ER, and LD was 97.64, 87.71, and 98.48%, respectively, and the SDs of the match ratios were all <10% (Table S2). Comparison of DeepContact segmentation to manual segmentation was also demonstrated by direct visualization (Fig. S2, a and b). DeepContact modeled the integrative patterns of ER, excluding other cisterna-like organelle networks from the model, such as the Golgi apparatus (Fig. S2 c). To further ensure segmentation of the genuine ER networks by DeepContact, HRP-KDEL diaminobenzidine (DAB) staining was used to highlight the ER structure in the EM picture. Segmentation of ER by DeepContact coincided mostly with DAB-enhanced HRP-KDEL ER networks (Fig. S2, e and f). In addition, the top likelihood and similarity loss implanted in DeepContact was effective in eliminating false positives for mitochondria (Fig. S1, b–e). These results manifest the accurate and stable predictions of DeepContact.

To demonstrate the efficiency of DeepContact, we recorded the time consumed by DeepContact analysis on a single Titan 1080Ti graphics processing unit core and compared it to the manual annotation by Labelme. DeepContact analysis can be divided into five stages: preprocessing, organelle #1 segmentation, organelle #2 segmentation, visualization, and MCS quantification. As expected, DeepContact exhibited significant advantages in the steps for segmenting various organelles when analyzing specific cultured cells or

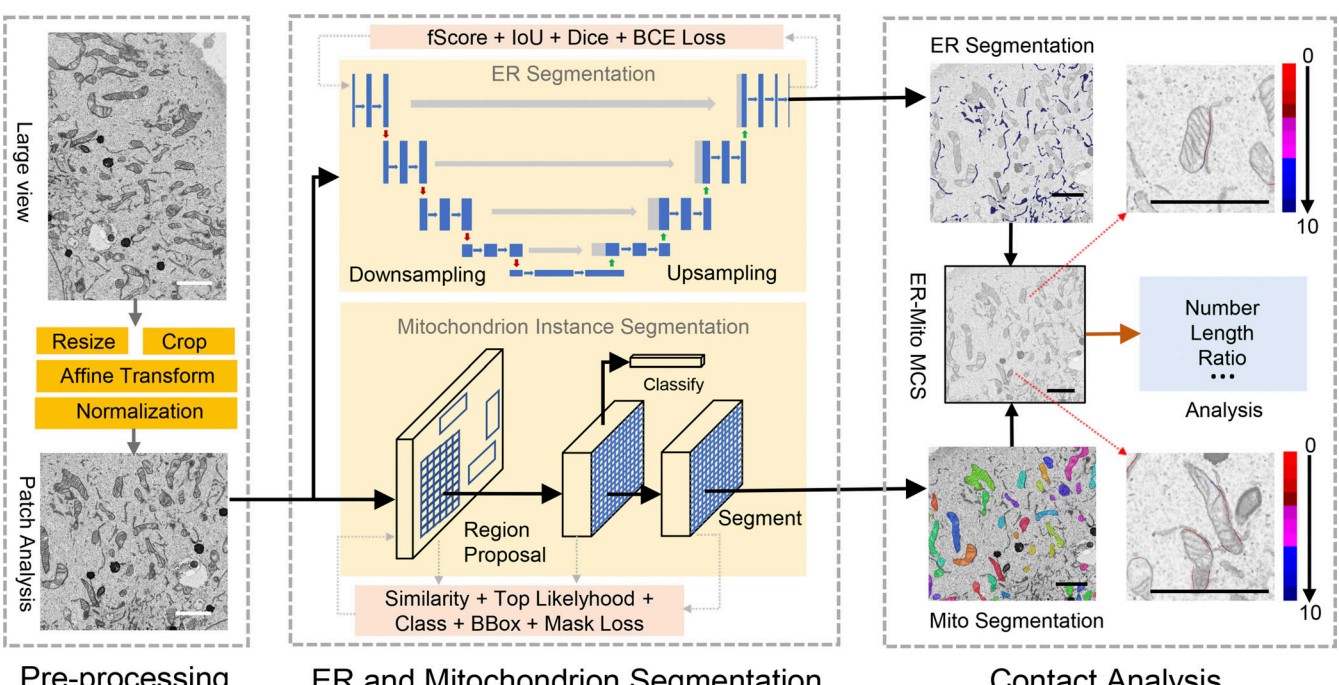

**Figure 1. The DeepContact workflow.** The large image was acquired from EM. We preprocessed this image by a series of operations, including resizing, cropping, affine transformation, and normalization to obtain the patch for analysis with a standard size of 1,024 × 1,024 pixels and a resolution of 10 nm. We adopted fScore loss, IoU loss, Dice loss, and BCE loss to train the ER segmentation model and similarity loss, top likelihood loss, CE loss, bounding box regression loss, and mask IoU loss to train the mitochondria segmentation model. The output of the ER segmentation model is the whole ER region. The output of the mitochondria segmentation model is all of the individual mitochondria instances. The number, perimeter, area, ratio, and MCS profile are calculated and reported. The MCS profile includes the lengths and ratios of 1–10-pixel-width intervals represented by different colors. Scale bar, 2 μm.

tissues (Tables S3, S4, and S5). In general, the program took seconds for each stage of DeepContact, whereas manual annotation lasted at least minutes, sometimes even hours. After segmentation, the calculation of contact parameters is even more complicated. An experienced researcher, usually with the help of software such as ImageJ, takes hours to process the equivalent amount of data (50 randomly selected patches). Importantly, the accuracy of the morphometric estimates by DeepContact was comparable to manual estimates (Fig. S2 g). These time contrasts indicate that, while ensuring accuracy, the performance of DeepContact is superior to manual annotation in regard to speed, making it highly feasible for high-throughput analysis.

### Optimization of the quantification procedure for MCSs by DeepContact

Quantitative analysis of MCSs has previously been attempted by using fluorescent probes at the contact. Therefore, we compared the commonly used ER-Mito contact indicator with Deep-Contact. When U-2 OS cells were transfected with either a rapamycin-induced ER-Mito tethering system (Csordás et al., 2010) or a split-GFP–based ER-Mito contact indicator (Yang et al., 2018), mitochondria degeneration (Fig. S3 b) or aggregation (Fig. S3 c) was readily detected by EM, raising concerns for MCS measurements when overexpression of split-GFP caused frequent mitochondrial abnormalities. In contrast, DeepContact requires no overexpression of probes. Mitochondrial degeneration or aggregation was rarely observed in wild-type U-2 OS

cells (Fig. S3 a). These results suggest that DeepContact is risk-free for artifacts introduced by probe overexpression.

ER-Mito MCSs in nutrition deprivation, when cells were cultured in HBSS, have been quantified using various Split-GFP indicators, including the short (∼10-nm) and long (10–50-nm) versions (Cieri et al., 2018). In these studies, an increased number of total MCSs were detected when using the short version, but not the long version, of the Split-GFP indicators (Cieri et al., 2018; Yang et al., 2018). Using the contact width setting in DeepContact, we found that both the number and length ratio of the ER-Mito contact were significantly increased in HBSS-treated cells in the short MCS width range (≤10 nm; Fig. S4, a and b). The number ratio of MCSs in the 10–50-nm range was not altered, which is consistent with the Split-GFP analysis (Fig. S4 a). However, the MCS length ratio measured by Deep-Contact was significantly increased in the 10–50-nm width range (Fig. S4 b), which was not seen in the previous analysis. These results suggest that a fluorescent probe–based analysis may report overall changes in MCS formation upon physiological shifts but falls short in picking up the details.

MCSs were shown to possess various width ranges according to the tethering conditions (Prinz, 2014; Scorrano et al., 2019). With DeepContact, changes in contact profiling could be dissected on pixel-based width intervals in 1-pixel steps, which corresponds to a length of 10 nm, from 0 to 10 pixels (0–100 nm). DeepContact was able to resolve these details based on accurate segmentation of the membrane boundaries of the ER and mitochondria (Fig. 2, a and b). In starved U-2 OS cells, we

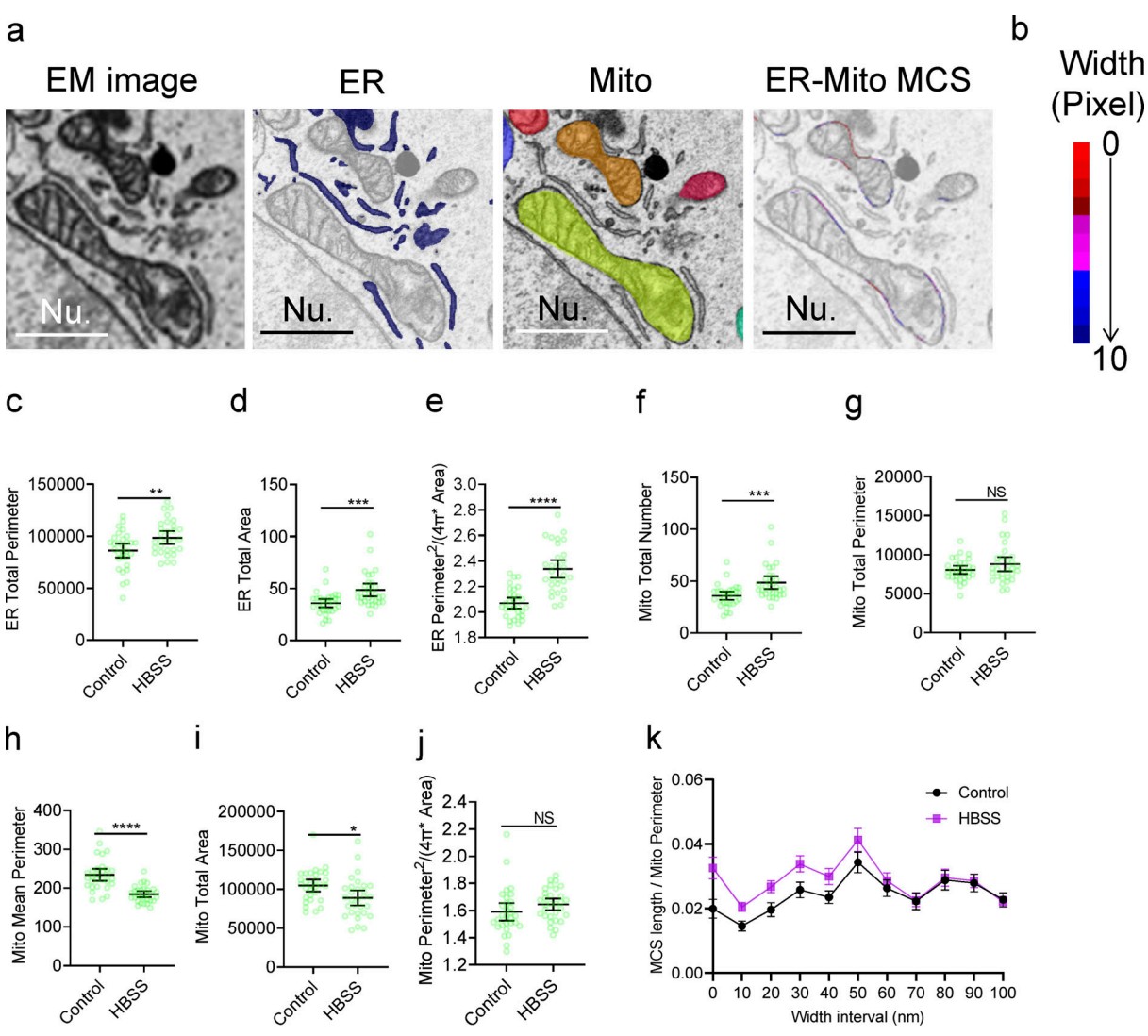

Figure 2. **Quantitative analysis of ER-Mito contact by DeepContact. (a)** Segmentation of the ER, mitochondria, and ER-Mito MCS as colored pixels on the outer mitochondria membrane spanning a series of width intervals from ER membranes. **(b)** The color bar indicates color gradients corresponding to 0–10 pixel-width (0–100 nm, 10-nm resolution) intervals of the MCS in panel a. Nu, nucleus. Scale bar, 1 μm. **(c)** Total ER perimeter in control and HBSS-treated U-2 OS cells. **(d)** Total ER area in control and HBSS-treated U-2 OS cells. **(e)** ER elongation condition in control and HBSS-treated U-2 OS cells. Elongation factor, perimeter$^2$/(4π × area). **(f)** Total number of mitochondria in control and HBSS-treated U-2 OS cells. **(g)** Total mitochondria perimeter in control and HBSS-treated U-2 OS cells. **(h)** Mean mitochondria perimeter in control and HBSS-treated U-2 OS cells. **(i)** Total mitochondria area in control and HBSS-treated U-2 OS cells. **(j)** Mitochondria elongation condition in control and HBSS-treated U-2 OS cells. Elongation factor, perimeter$^2$/(4π × area). **(k)** Ratio of MCS length/mitochondria perimeter in 0- to 100-nm-width intervals with HBSS treatment. The sample size of each experimental setting is 30, and individual dots in the plot represent the mean value of a 15 × 10–μm cellular image. Bars in c–k indicate 95% confidence intervals.

noted an increase in the total perimeter of the ER (Fig. 2 c), ER areas (Fig. 2 d), and ER elongation (Fig. 2 e), an increase in the number of mitochondria (Fig. 2 f), and a more prominent decrease in the mean mitochondrial perimeter (Fig. 2 h) compared with control cells. In addition, the MCS length ratio was constantly increased in the width interval of 0–60 nm but became the same when the width range was wider (Fig. 2 k). Thus, DeepContact is capable of gathering comprehensive contact-related measurements, including organelle number, perimeter, area, and morphological geometry (Fig. 2, c–j), as well as pixel-based width counting (Fig. 2 k), with previously unachieved precision.

Next, we tested the sensitivity of DeepContact in settings with more physiological relevance. Autophagic protein EPG-3/VMP1 has been shown to regulate ER contact with numerous organelles, including mitochondria (Zhao et al., 2017). When deleted in cells, the ER-Mito contact is evidently increased. In contrast, contact site adapter protein VAPa/b is well established as mediating a variety of ER-based contacts. In the case of ER-Mito contact, PTPIP51 or Vps13D bridges the two organelles in a VAP-dependent manner (Guillén-Samander et al., 2021). As a control, atlastin (ATL) is an ER membrane fusogen, the lack of which causes an aberrant ER morphology but with no known impact on ER-Mito contact. Thus, we analyzed VMP1-deleted

cells, VAPa/b-depleted cells, and ATL-deleted cells using Deep-Contact. As expected, DeepContact obtained consistent results (Fig. S4, c–f) with the functional scenario of these molecular machineries and the published results obtained with manual segmentations (Zhao et al., 2017). These results confirm that DeepContact is capable of detecting changes in MCS profiling in physiological settings.

EM sample preparations of adherent cells were used to maintain the native topology of the intracellular organizations, as well as the morphological status of the organelles. However, section z-axis levels from the adherent side (equivalent to the basal side of polarized cells) to the top side (equivalent to the apical side) may possess different properties in MCS organization. To test this hypothesis, we acquired image datasets from three sections, each with an interval of ~2.5 µm in the z-axis, within serial sections of adherent cells on sapphire disks. The ER-Mito contact length ratio showed no significant differences between sections (Fig. S4 i). However, the total ER perimeter (Fig. S4 g) and mean mitochondrial perimeter (Fig. S4 h) were slightly different in apical sections compared with the basal and middle sections. These results indicate that, for unbiased analysis, comparisons between samples from a similar section level are highly recommended.

We incorporated an active learning framework (Yang et al., 2017) to reduce the annotation effort by making judicious suggestion on the most effective annotation samples (Fig. 3 a). Starting as a set of a certain initial number of labeled data, we iteratively trained our model. During each stage, a new model was trained and tested. We selected the worst 10% of images based on the mIoU value, from which we identified organelles that are missed in prediction as difficult cases. For additional images to be added in the next round of training, we ensured that the newly unannotated images contained sufficient "difficult" cases with similar features so that the model could be improved. After acquiring the new annotation data, we started the next round of training using all available annotated images. Taking drug-induced morphological alterations of mitochondria as an example, we efficiently incorporated aberrant morphologies into the mitochondria models via targeted labeling and training, and improvement of the model was readily verified by visualization (Fig. 3 b). As a result, we noticed that three mitochondrial drugs, carbonyl cyanidem-chlorophenyl hydrazone (CCCP), oligomycin A1, and mdivi-1, all decreased the ER-Mito contact ratios in all width interval ranges (Fig. 3 c).

## MCS profiling under combined nutrient conditions

Next, we applied DeepContact to a biological setting in which changes in the MCS profile are important and physiologically relevant. Contact between organelles is actively involved in nutrient homeostasis (Lahiri et al., 2015; Prinz et al., 2019). In addition to the ER-Mito contact extensively analyzed above, contact between LDs and mitochondria is essential in regulating lipid metabolism and energy homeostasis (Benador et al., 2019; Lahiri et al., 2015). To investigate MCS coordination in adapting to a varied nutritional supply, we performed DeepContact analysis in U-2 OS cells under three different nutrient conditions, including FBS starvation for the depletion of lipid

resources, glucose and sodium pyruvate starvation, and HBSS starvation limiting both lipid and sugar intake. Segmentation of the ER, mitochondria, and LD was performed simultaneously by DeepContact, and the data were analyzed for both the ER-Mito and LD-Mito contacts (Fig. 4, a and b). The dimensions and shapes of the ER and mitochondria were altered incidentally, and no consistency was observed between starvation treatments (Fig. S4, j–l). LD expansion was readily detected in cells during glucose starvation or HBSS treatment, but not in FBS-starved cells (Fig. 4 c). LD-Mito contact is suggested to be within 30 nm in width (Lahiri et al., 2015); therefore, ≤30 nm contacts were summed for LD-Mito contact analysis. Both the ER-Mito contact in short width intervals (≤10 nm) and the LD-Mito contact were generally increased under these conditions compared with untreated cells (Fig. 4, f, h, i, and k). ER-Mito contact with a width interval ranging from 20 to 80 nm was slightly decreased when serum or glucose starvation was applied (Fig. 4, i and l). We also introduced overfeeding of oleic acid (OA) either before or during starvation to monitor contact adaptation. Consistently, the growth of LDs was evident upon OA treatment (Fig. 4, d and e). Surprisingly, even though LD-Mito contacts were further induced before and during OA conditions, ER-Mito contacts were generally reduced, particularly in the serum or glucose starvation groups compared with cells with no OA treatment (Fig. 4, f–l). These results indicate previously unidentified metabolic coordination between ER-Mito and LD-Mito contacts and demonstrate the ability of DeepContact to reveal systemic changes in multiple types of MCSs in a comprehensive manner.

## MCS profiling in epithelial tissues

Quantitative analysis of MCSs in tissues is challenging due to variations in cell type and complicated cell–cell entanglement. At the same time, it is urgently needed because MCS organization is tightly linked to physiological or pathological processes. Therefore, we developed a DeepContact tissue model for cell type–specific MCS analysis. Sertoli cells are the only somatic cell type in the seminiferous epithelium, which provides structural support and nourishment for spermatogenesis (Hess and Vogl, 2015). Surrounded by Sertoli cells, germ cells develop in a cyclic manner during migration from the basal to the adluminal site of the seminiferous epithelium (Hess and Renato de Franca, 2008; Leblond and Clermont, 1952). Organelle communication within Sertoli cells has been suggested to play a role in the progression of spermatogenesis throughout the seminiferous epithelial cycle (Lyon et al., 2017; Vogl et al., 2018). We examined whether the dynamics of the ER-Mito contact in these cells fit with the cycle. Sertoli cells possess an enormous territory with basal-apical polarities (Hess and Vogl, 2015). For the reasons discussed above, we focused on the basal sections of these cells for unbiased ER-Mito contact probing by DeepContact (Fig. S5). The seminiferous epithelium cycle of the mouse has been classified into 12 stages, which are roughly grouped into early, middle, and late stages (Cheng, 2009). Both the total ER perimeter and mean mitochondrial perimeter were similar between the early and middle stages, and both were increased significantly in the late stages (Fig. 5, a and c). In contrast, the elongation factor for both the ER and mitochondria had an apparent decrease in the late

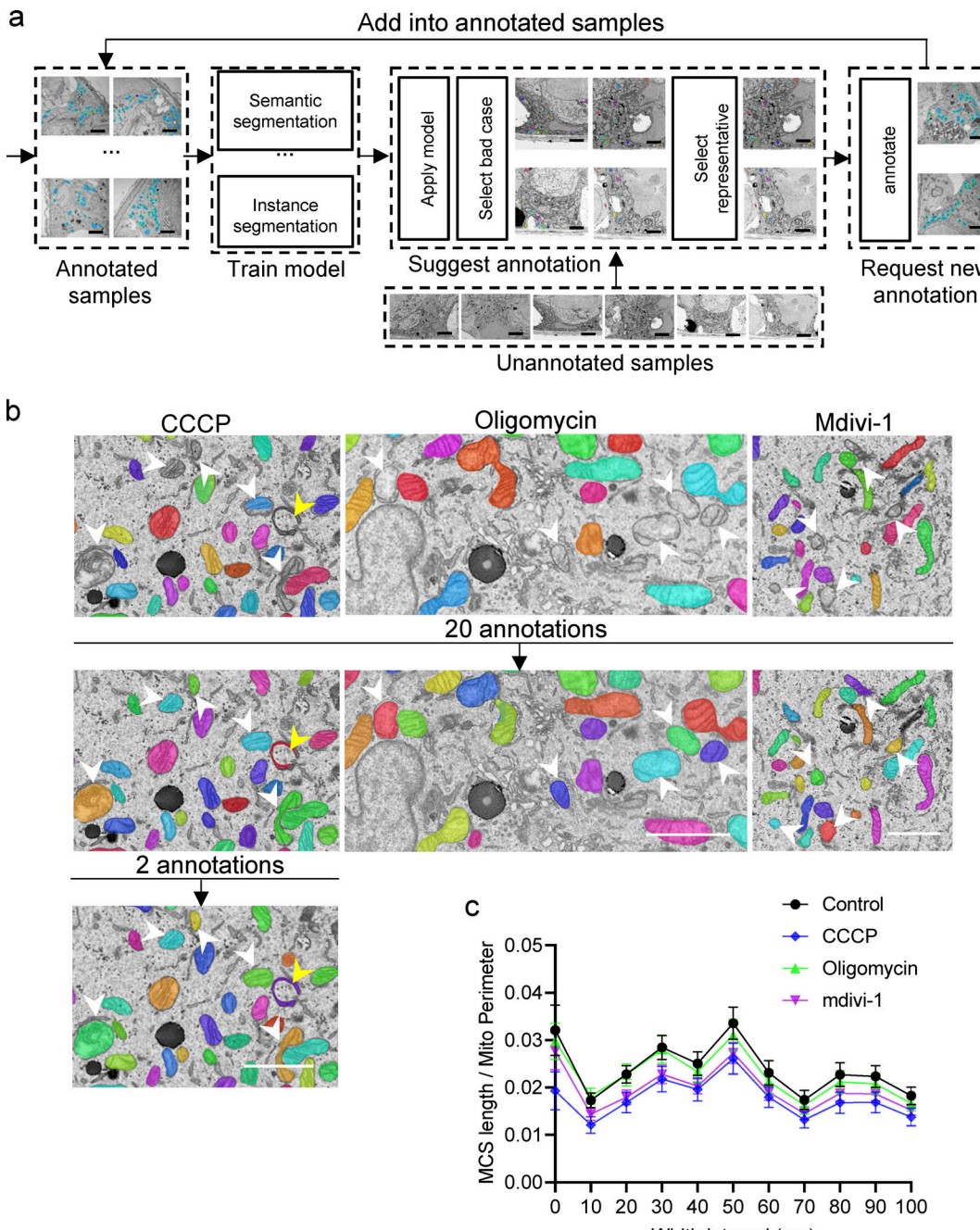

Figure 3. **Incorporation of new morphologies in the mitochondria model via the active learning function of DeepContact. (a)** Demonstration of the active learning procedure. **(b)** Incorporation of new morphologies induced by 4 h of treatment with 10 µM CCCP, 10 µg/ml oligomycin A1, and 50 µM mdivi-1 through the active learning function of DeepContact. Scale bar, 2 µm. **(c)** MCS length ratio in 0- to 100-nm-width intervals in control, CCCP, oligomycin, and mdivi-1 treatments. Arrowheads in b indicate mitochondria that are not segmented by the original model but are segmented by the refined model through two rounds of active-learning process. The sample size of each experimental setting is 30, and individual dots in the plot represent the mean value of a 15 × 10–µm cellular image. Bars in c indicate 95% confidence intervals.

stages (Fig. 5, b and d). Furthermore, the ER-Mito contact length ratio in ≤30-nm-width intervals appeared to be lower in the late stage than the early and middle stages. No substantial difference was observed between stages when the width intervals were ≥40 nm (Fig. 5 e). Next, we analyzed ER-Mito contact profiling using width intervals ≤30 nm in seminiferous tubule samples with adequate staging features in a large view of the ultrathin section, so that corresponding stages could be determined accurately (Fig. S5). The contact increased gradually from stage I, reaching the highest level at late stage VII, before decreasing a little at late stage VIII, but then decreased significantly at stage IX and was sustained at a low level until stage XII (Fig. 5 f). Notably, cisternal ER alignment with elongated mitochondria was observed in most cases in late stage VII (Fig. 5 g), whereas

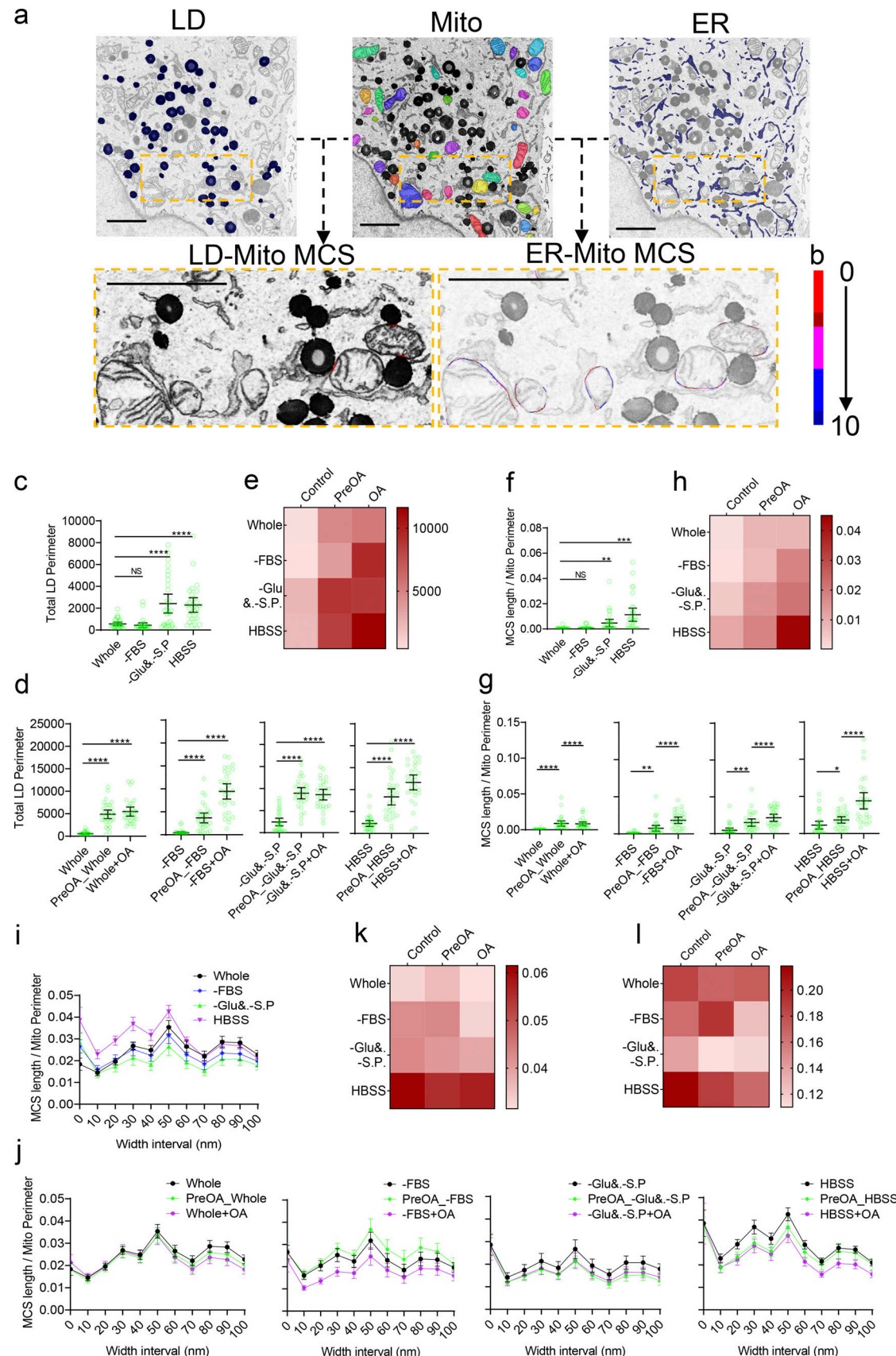

Figure 4. **MCS coordinations in various nutrient conditions. (a)** Segmentation of LD, mitochondria, ER, LD-Mito MCS, and ER-Mito MCS by DeepContact. Scale bar, 2 µm. **(b)** The color bar indicates color gradients corresponding to 0- to 100-nm-width intervals of the ER-Mito MCS in panel a. **(c)** Total LD perimeter

in complete medium (Whole), FBS starvation (–FBS), glucose and sodium pyruvate starvation (–Glu&. –S.P.), and HBSS starvation conditions. **(d)** Total LD perimeter with OA treatment 10 h before (Pre-OA) or during (OA) starvation treatment in Whole, –FBS, –Glu&. –S.P., and HBSS conditions. Units in c and d are pixels. **(e)** Levels of total LD perimeter in Whole, –FBS, –Glu&. –S.P., and HBSS conditions combined with Pre-OA or OA treatments. **(f)** Length ratio of <30-nm LD-Mito MCS in Whole, –FBS, –Glu&. –S.P., and HBSS conditions. **(g)** Length ratio of <30-nm LD-Mito MCS with Pre-OA or OA treatment in Whole, –FBS, –Glu&. –S.P., and HBSS conditions. **(h)** Levels of LD-Mito MCS in Whole, –FBS, –Glu&. –S.P., and HBSS conditions combined with Pre-OA or OA treatments. **(i)** ER-Mito MCS length ratio in 0- to 100-nm-width intervals in Whole, –FBS, –Glu&. –S.P., and HBSS conditions. **(j)** MCS length ratio in 0- to 100-nm-width intervals with Pre-OA or OA treatment in Whole, –FBS, –Glu&. –S.P., and HBSS conditions. **(k and l)** Levels of <10-nm (k) and 20–80-nm (l) ER-Mito MCS length ratios in Whole, –FBS, –Glu&. –S.P., and HBSS conditions combined with Pre-OA or OA treatments. The sample size of each experimental setting is 30, and individual dots in the plot represent the mean value of a 15 × 10–μm cellular image. Bars in c, d, f, g, i, and j indicate 95% confidence intervals.

swelled circular ER structures often touched round mitochondria in stage IX (Fig. 5 h). Taken together, these results of the DeepContact tissue model reveal a wave of ER-Mito contact changes in Sertoli cells, accompanied by morphological shifts in the participating organelles, which are likely tightly associated with functional switching in the cell.

## Discussion

To systematically quantify intracellular MCS profiles using EM images, we designed DeepContact and tailored it for high-throughput capacity. The accuracy and efficiency of the procedure is ensured by following modifications to the conventional workflow. First, EM samples can be prepared by the reduced osmium-thiocarbohydrazide-osmium (ROTO) method, which highlights the organelle outline by preferential staining of lipid bilayers and subsequently benefits the recognition of organellar boundaries. Second, the active learning procedure allows human intervention by interactive visual checking and refining the models by targeted labeling, which then ensures the incorporation of new organelle morphologies in an accurate and efficient manner. Along these lines, top likelihood sampling combined with similarity loss is applied in segmentation models to avoid the potential risk of false positives. Finally, MCS measurements are plotted with pixel-based width intervals, which provides full-scale information on the contact, including features of the tethering conditions. Such adjustments enable counting not only the number of MCSs, but also accurate width-based lengths of each MCS, as the width of an MCS is a key structural element for its function and is tightly regulated in the cell (Giacomello and Pellegrini, 2016).

EM image–based quantification by DeepContact bypasses the need for the fluorescent indicators commonly used in optical microscopy–based MCS analysis. As revealed here, and consistent with previous reports, overexpression of these indicators, including Split-GFP, induces artificial tethering of corresponding organelles, which results in abnormal organelle morphologies (Kakimoto et al., 2018; Tashiro et al., 2020). In addition, contact analysis based on fluorescence colocalization is limited by optical resolution, and it is usually challenging to introduce the indicators when tissue samples are used instead of cultured cells. Therefore, DeepContact offers unprecedented precision in analyzing a variety of MCSs in cultured cells or tissue samples.

Using DeepContact, we were able to reveal subtle specific changes in MCS profiles with cells under different nutrient conditions. In particular, we found that the length ratio of both LD-Mito and ER-Mito MCSs in short-range (≤10-nm) width

intervals are mostly increased in nutrient-depleted conditions, and excessive fatty acids, in the form of OA treatment, caused a further increase in LD-Mito contact but a decrease in ER-Mito contact. These findings support cross-talk between LD-Mito and ER-Mito contacts in regulating lipid and energy homeostasis. In a simple course of starvation, especially glucose starvation or HBSS-induced starvation, intracellular lipids are quickly gathered by either elevated intake or redistribution through autophagy and flooded into LDs. A balance is then reached with help from both LD-Mito and ER-Mito contacts. Interestingly, when such a balance is once again challenged by a sudden supply of fatty acids, LD-Mito contact appears to become dominant in handling the response, whereas ER-Mito contact likely makes way for this. A mechanistic investigation of these changes warrants a further understanding of the metabolic pathway.

We also took advantage of DeepContact by probing the MCS profiles in an epithelial tissue model. Fully differentiated Sertoli cells adapt subcellular reorganization along with the procession of the seminiferous epithelial cycle and spermatogenesis (Ueno and Mori, 1990). Morphometric studies of subcellular organelles were performed extensively prior to the 1990s, revealing cyclic alterations of several organelles in number or area parallel with the seminiferous epithelial cycle, including mitochondria, ER, Golgi apparatus, and primary and secondary lysosomes (Kerr, 1988; Morales et al., 1986; Ueno and Mori, 1990). With DeepContact, we found that ER-Mito contact changes in a wave-like pattern in mouse Sertoli cells. The wave peaks in late stage VII and quickly bottoms out in stage IX. Coincidently, in stage VIII, which is the critical switching point in the wave, two major events in the process of spermatogenesis occur concurrently: mature sperm is released to the lumen of the seminiferous tubule, and the blood–testis barrier is reconstructed to facilitate entry of the preleptotene primary spermatocyte into the immune-privileged luminal compartment from the basal compartment. Importantly, both events are orchestrated by Sertoli cells (Cheng, 2009). It is reasonable to speculate that ER-Mito contact rearrangements play a key role in spermatogenesis. Similarly, in the late stages of spermatogenesis, from stage IX to stage XI, primary spermatocytes proceed in prophase differentiation steps, and the only generation of spermatids gradually elongate and develop more compacted nuclei; however, no new associations of germ cells with the Sertoli cell occur in such late stages of the seminiferous epithelial cycle. Two successive meiotic divisions occur at stage XII, and the highest numbers of spermatid are reached at stages I–II, which require tremendous new associations with Sertoli cells. Such changes may demand more actively integrated functions of Sertoli cells, which are

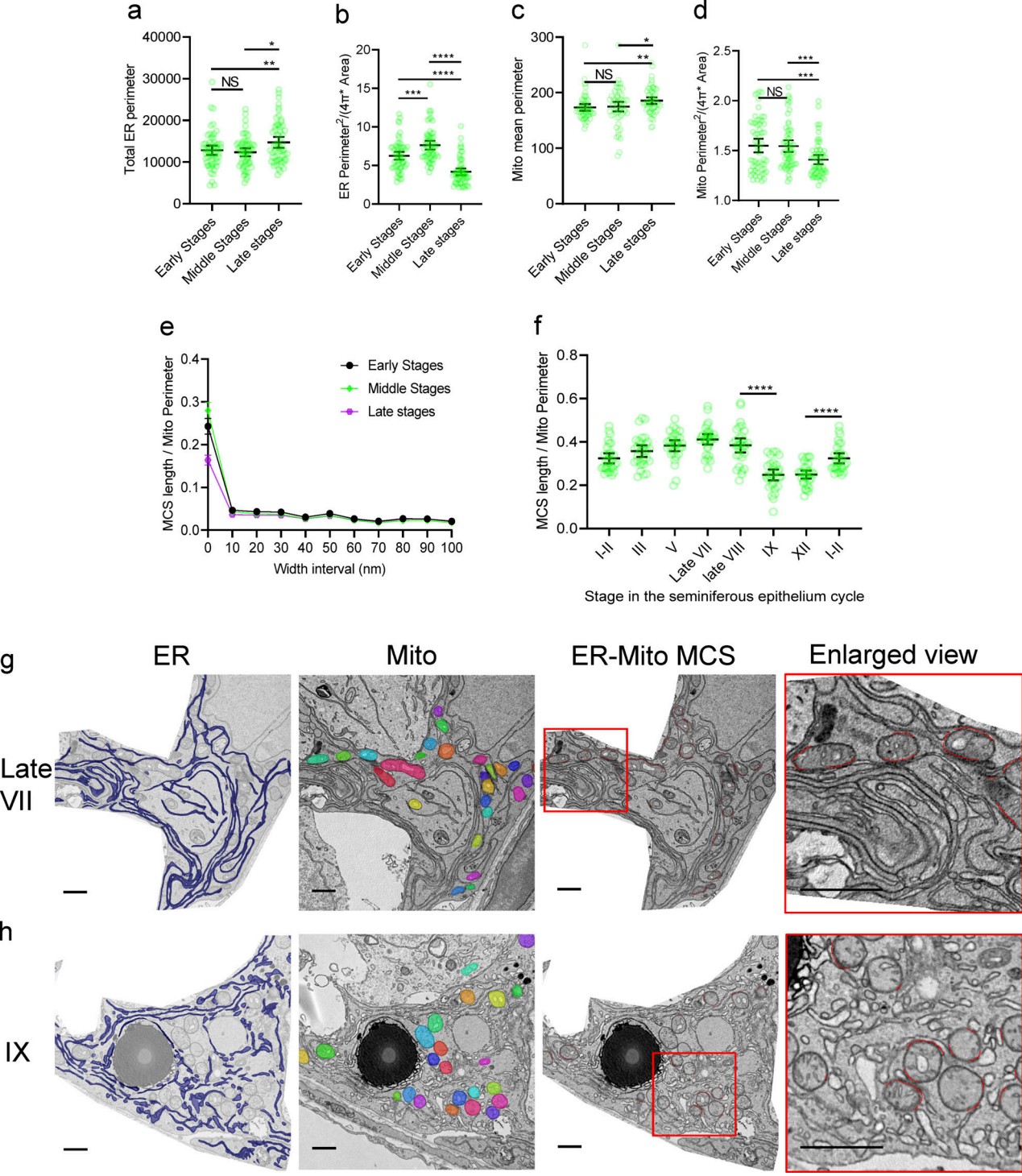

Figure 5. **MCS wave in Sertoli cells of seminiferous epithelial tissue. (a–e)** Total ER perimeter (a), ER elongation condition (b), mean mitochondrial perimeter (c), mitochondria elongation condition (d), and MCS length ratio in 0- to 100-nm-width intervals (e) in early, middle, and late stages of the seminiferous epithelium cycle. $n$ = 60; 15 images from four seminiferous epithelial tubules were included in each sampling. **(f)** ER-Mito MCS length ratio in representative stages of the mouse epithelium cycle. Significance was calculated between neighboring stages. Data set from stage I–II was arranged in both the beginning and the last column of the plot for a statistical comparison between neighboring stages. $n$ = 30; 15 images from two seminiferous epithelial tubules were included in the sampling of each stage. Individual dots in the plots in a, b, d, and f represent the mean value of a Sertoli cell in the basal area of the seminiferous tubule (15 × 10–μm size). Significance was calculated by unpaired $t$ test with Welsh's correction. *, P < 0.05; **, P < 0.01; ***, P < 0.001; ****, P < 0.0001 with 95% confidence intervals. **(g and h)** Segmentation of ER, mitochondria, ER-Mito MCS, and enlarged view of ER-Mito MCS in ≤30-nm-width intervals of the Sertoli cells from the basal area of a seminiferous tubule in late stage VII (g) and stage IX (h). Bar, 1 μm. Individual dots in the plot represent the mean value of a 15 × 10–μm cellular image. Bars in a–f indicate 95% confidence intervals.

achieved by increased levels of ER-Mito associations. As expected, the overall levels of ER-Mito contact quantified by DeepContact appear to correlate well with the number of spermatids associated with the Sertoli cell and major dynamics of spermatogenesis. These findings collectively confirm the power of DeepContact.

## Materials and methods

### Cell culture
U-2 OS (ATCC) cells were cultured in DMEM supplemented with 10% FBS and 1% penicillin-streptomycin at 37°C in a 5% CO$_2$ atmosphere. Three starvation conditions were applied by depleting nutrients from the complete medium for 12 h, including FBS deprivation, glucose and sodium pyruvate deprivation, and depletion of all energy substrates using HBSS. OA was supplied at 200 μM for 12 h either before or during the starvation treatment.

All drugs were obtained from Sigma-Aldrich. CCCP, mdivi-1, and oligomycin A1 were used to induce mitochondrial abnormalities. We applied 10 μM CCCP, 50 μM mdivi-1, and 10 μg/ml oligomycin A1 in complete medium for 4 h. Standard procedures of Lipofectamine 3000 (Thermo Fisher Scientific) transfection were used to transfect pDisplay-HRP-KDEL (#85582; Addgene) or pDisplay hollow vector (V66020; Invitrogen) to the cells.

### HPR staining for EM
DAB staining was performed according to previously reported procedures (Galmes et al., 2016; Shi et al., 2017) with slight modifications. Briefly, 1.25% glutaraldehyde in PBS was used to fix the cell monolayer for 1 h on ice. Fixatives were neutralized and washed with 0.1 M ammonium phosphate, pH 7.4, for 5 min three times, 20 mM glycine in PBS for 5 min, and PBS for 2 min three times on ice. DAB Kit (CWBIO) was used to stain cells for 20 min on ice, followed by three 2-min rinses in chilled PBS. The formation of DAB polymer could be checked by transmission light microscopy. Stained cells could be stored in 2.5% glutaraldehyde in a refrigerator if not proceeding directly to EM sample preparation processes.

### EM sample preparation of cultured cells
EM sample preparations of adherent cells were used for all cultured cells to keep the topology of the subcellular organizations. Sapphire disks (Wulundes, 3 × 0.16 mm) were placed on the bottom of a 30-mm Petri dish and coated with 0.02% poly-lysine for 30 min, and then washed twice with culture medium after withdrawing the coating medium. U-2 OS cells were seeded on the Petri dish with the sapphire disk and allowed to reach ~80% confluence before fixation. Adherent conditions of the cell monolayers on the sapphire disk were maintained through the whole EM sample preparation procedure to retain the physiological topology of the subcellular structures. ROTO was used to enhance the contrast of the lipid-based membranous structures (Tapia et al., 2012) with slight modifications. Briefly, cells were fixed with 2.5% (vol/vol) glutaraldehyde in phosphate buffer (0.1 M, pH 7.4) and washed four times in phosphate buffer, then immersed in 1% (wt/vol) OsO$_4$ and 1.5% (wt/vol) potassium ferricyanide aqueous solution at 4°C for 30 min. After washing,

cells were incubated in filtered 1% thiocarbohydrazide (Sigma-Aldrich) solution at room temperature for 15 min, and then with 1% unbuffered OsO$_4$ at 4°C for 30 min and 2% uranyl acetate at room temperature for 1 h, followed by four rinses in ddH$_2$O for 6 min each between each step. Cells were then dehydrated through graded ethanol (30, 50, 70, 80, 90%, and 2 × 100%, 5 min each) into pure acetone (2 × 5 min). Dehydrated samples were infiltrated in graded mixtures (3:1, 1:1, and 1:3) of acetone with SPI-PON812 resin (21 ml SPI-PON812, 13 ml dodecenyl succinic anhydride, and 11 ml nadic methyl anhydride), and then pure resin. Finally, a sapphire disk with a cell monolayer facing upward was embedded in the bottom of a tubule mold (GP130-25; EMCN) filled with pure resin containing 1.5% BDMA accelerator. Resin samples were polymerized for 48 h at 60°C. The sapphire disk was removed after a brief cold shock of the resin sample blocks with liquid nitrogen.

### Animals
10-wk-old male CD-1 mice were housed in groups of three to five at 22–24°C with a 12-h light/dark cycle. Animals had access to water and food ad libitum. All experiments were approved by the Animal Care Committee at the Institute of Biophysics (license no. SYXK2016-19).

### EM sample preparation of seminiferous tissues
The testis was exposed in the lower abdomen after anesthetizing the mice via isoflurane inhalation. A pair of testes were transpierced with a 26-gauge needle in the middle shaft of the short axis and injected slowly with 0.2 ml first fixative (2% PFA + 2.5% glutaraldehyde in PBS) from one end of the long axis for brief fixation of the seminiferous tissues under physiological conditions. The testes were then dissected out and immersed into fresh first fixative. The tunica albuginea was carefully removed using ophthalmic scissors, and bundles of seminiferous tissue with <20 seminiferous tubules were gently dissected using the ophthalmic scalpel and further fixed in first fixative with gentle shaking for 4 h under 4°C refrigerator. ROTO procedures were adapted for the seminiferous tissue sample preparation. Dissected tissue was first immersed in 1% (wt/vol) OsO$_4$ and 1.5% (wt/vol) buffered with 0.1 M cacodylate (pH 7.4) at 4°C for 90 min. Tissues were incubated in filtered 1% thiocarbohydrazide at room temperature for 30 min, 1% unbuffered OsO$_4$ aqueous solution at 4°C for 90 min, and 2% uranyl acetate aqueous solution at 4°C overnight. Triple rinses were performed in ddH$_2$O for 30 min between steps. The next day, progressive lowering temperature dehydration was performed in the freeze substitution device (AFS2; Leica) to reduce extraction-induced changes to the morphology of the membranous structures (Carlemalm et al., 1985). After warming up the dehydrated sample to room temperature in 100% ethanol, the ethanol was exchanged with 100% acetone twice. Dehydrated samples were infiltrated in graded mixtures (3:1, 1:1, and 1:3) of acetone with SPI-PON812 resin, then pure resin for 2 d with exchange every 12 h. Seminiferous tissue bundles were embedded in pure resin with 1.5% BDMA accelerator and aligned longitudinally with the wells of embedding mold and polymerized in an oven at 45°C for 24 h and 60°C for 48 h.

## Ultrathin sectioning

ROTO-prepared samples of both U-2 OS cells and seminiferous tissue were cut into ultrathin sections 70-nm-thick using a microtome (EM UC6; Leica), collected with PE tape mixed with carbon nanoparticles, and then mounted on a wafer as support (Li et al., 2017). Serial ultra-thin sections of sapphire disk–assisted adherent U-2 OS cells were cut by AutoCUTS (Li et al., 2017) to compare membrane contact conditions among sections from the basal, middle, and apical Z-level.

## Scanning field emission EM data acquisition

For U-2 OS cells, 3,072 × 2,048 images at 5-nm resolution were acquired in the main cytoplasmic area near the nuclei using a circular backscatter (FEI Helios Nanolab 600i dual-beam scanning EM) detector under immersion high-magnification mode (2-kV accelerating voltage, 0.34-nA beam current, 2-μs dwell time, and 400-V reverse bias voltage). For seminiferous samples, a map view of a whole seminiferous tubule was acquired to stage the seminiferous cycle according to the cellular associations between the germ cells and Sertoli cells, mitosis/meiosis status of the germ cells, presence of one or two generations of spermatids, and the differentiation status of spermatids (Hess and Renato de Franca, 2008; Oakberg, 1956). The latter includes the perinucleus formation of the acrosomal granules, capping of the acrosomes, formation of the tail and cytoplasmic residual bodies, and condensation of the chromatin of the spermatid (Hess and Renato de Franca, 2008). Stages VII, VIII, and IX are those before, during, and after spermiation, respectively. Stage XII and I–II correspond to those during and after the meiotic divisions of the spermatocytes, respectively. Sertoli cells are irregularly shaped, spanning an ~100-μm range from the very basal to the adluminal compartment of the seminiferous epithelium. Targeted 1,024 × 1,526 (Fig. S5) images at 10-nm resolution were acquired in the basal region of the Sertoli cell to avoid basal-apical polarity-induced variance using a circular backscatter (FEI Helios Nanolab 600i dual-beam scanning EM) detector under immersion high magnification mode (2-kV accelerating voltage, 0.69-nA beam current, 5-μs dwell time, and 400-V reverse bias voltage).

## Transmission EM data acquisition

ROTO-prepared Cos7 cells with EPG-3/VMP1 deletion, VAPa/b depletion, and ATL deletion kindly provided by other researchers (Zhao et al., 2017) were cut into 70-nm-thick sections and examined under a transmission electron microscope (FEI Tecnai Spirit120 kV) equipped with Morada G3 (EMsis) at 6,800× with 4.68-nm resolution under 100-kV accelerating voltage.

## Data annotation

Labelme software (Torralba et al., 2010) was used for manual annotation and segmentation. For cultured cell mitochondria, ER and LDs are annotated on 5-nm-resolution EM images. Approximately 60 EM images of U-2 OS cells and an additional 17 EM images of TM4 cells (a Sertoli cell line) were annotated to train the mitochondria model. Four EM images of TM4 cells and

12 EM images of U-2 OS cells were annotated to train the ER model. Approximately 20 EM images of U-2 OS cells in normal culture conditions or with LD induction by OA treatment and HBSS treatment were annotated to train the LD model. For Sertoli cells, we first annotated the plasma membrane in the images of the seminiferous epithelial tissue, which was then used to extract the RoI to exclude the cellular areas of various kinds of germ cells and peripheral smooth muscle cells. The mitochondria and ER were elaborately annotated by Labelme in ~50 selected representative images from stage I–II, stage III, late stage VII, stage IX, and stage XII to initialize the DeepContact model. The annotated data were randomly split in a 3:1 ratio to form the training: validation set. After training the initially labeled dataset, we evaluated the performance and added the training set as needed. An active learning framework was used to reduce the annotation effort by making judicious suggestions on the most effective annotation samples. Details of the active learning framework can be found in Fig. 3.

## Top likelihood sampling

Cellular structures possessing similar morphological features as the targeted organelle may cause false positives in organelle segmentation in EM images. Here, we propose a top likelihood sampling strategy incorporating the top likelihood loss (Xiao et al., 2019) in the Mask R-CNN (He et al., 2020) model, which samples the most suspected target regions during training. The top likelihood loss selects the top scoring negative anchors on the Region Proposal Network (RPN) and optimizes them. The anchors with top scores should be more representative of the suspected target regions as the training progresses. On the other hand, as long as these top scoring anchors are minimized, all anchors are simultaneously optimized toward negative. The RPN loss with the top likelihood sampling is expressed as

$$L_{tploss} = \frac{1}{N_{pos}} \sum_{i \in (\text{lowest} p_i)} \left[ L_{cls}(p_i, p_i^* = 1) + \lambda L_{reg}(t_i, t_i^*) \right],$$

$$L_{tnloss} = \frac{1}{N_{neg}} \sum_{i \in (\text{top} p_i)} L_{cls}(p_i, p_i^* = 0).$$

A similarity loss is then added to further identify the selected negative proposals from positive proposals. Fig. S1 a shows the detailed framework for implementing the top likelihood sampling with similarity loss. Fig. S1, b–e, shows the effectiveness of such implementation.

## Training details

For the Mask R-CNN model, the threshold score of detection as a mask is set as the default value of 0.5. The backbone network was Resnet101, and we used the stochastic gradient descent optimization method. The learning rate was set to 0.001 with a weight decay of 0.001. We first iterated 15 epochs to train the network head and iterate another 10 epochs to train all layers. The overall training procedure was continued for ≤25 epochs until the loss reached equilibrium. The learning rate of the U-Net model was set at 0.0005 with a weight decay of 0.0003. The training procedure for U-Net lasted for 130 epochs until it reached equilibrium.

## Deep-learning analysis procedure

For cultured cells, all samples are selected for the analysis (Fig. 1). For each sample, the well-trained DeepContact model was then used to segment each mitochondrion instance and the ER region. For Sertoli cells, DeepContact extracts the RoI, which is the area within the Sertoli cell territory determined by annotation of the plasma membrane. The IoU, which represents the ratio of the RoI area in the whole image, is computed for each sample. Only samples with IoU > 0.15 are selected for analysis (Fig. S5). Next, DeepContact extracts the boundary of each mitochondrion to compute the ER-Mito contact according to the distances between the boundaries of the mitochondria and ER region. Finally, the ER-Mito MCS length ratio is computed as

$$\text{Ratio} = \text{mito\_boundary}_{contact} / \text{mito\_boundary},$$

where mito_boundary refers to the length of all boundaries of the mitochondria among the patches generated from one EM image, and mito_boundary$_{contact}$ refers to the length of all contact between the mitochondria and ER among the patches generated from one EM image.

## Visualization and quantitative analysis of MCS distributions in different length intervals

To plot the MCS distribution with pixel-based width intervals among the contact boundary, we quantified each pixel's contact length on the edge of the organelle. Biologically, the contact satisfies one-to-one mapping; namely, each site on one organelle can communicate only with a unique site on another organelle. However, the size of some sites is only 1–2 nm, whereas the resolution of an analyzed image is 10 nm, limiting the application of this principle in the analysis. Nevertheless, we chose a tradeoff strategy that a pair of pixels (sites) has contact if there is no other site that has a distance 1 pixel (>10 nm) shorter than the distance of the pairs. Taking the ER-Mito MCS as an example, the procedure for quantifying the contact length ratio is as follows: define a value threshold; a pixel on the mitochondria edge is considered to contact a pixel on the ER if they have a distance less than or equal to the threshold (i.e., 10 pixels); extract the edge of the ER (Edgeer) and edge of the mitochondria (Edgemito) separately; find the pixels on the mitochondria that overlap with the ER (these pixels will not be taken into consideration for contact analysis); define the matrix MinDistance, which represents the minimum distance on each ER edge with at least one pixel on the mitochondria edge within the distance; search the pixels within the threshold on the mitochondria edge; ignore the ER edge if its distance from the mitochondria edge is 1 pixel greater than its MinDistance value; find the minimum distance that the ER edge has contact with the mitochondria edge and record the value in DistanceMap; and define the mitochondria edges that overlap with ER areas with DistanceMap values of 0.

The pseudo-algorithm is summarized as follows: Algorithm 1 Calculate Contacting

Require: Mito: Prediction of mitochondria; ER: Prediction of ER; threshold: the maximum distance which considers as a contact.

Ensure: nmito, ncon: Number of mitochondria and mitochondria-ER contact;

lenmito, lencon: Length of mitochondria and mitochondria-ER contact;

MinDistance: Distance on ER edge between edge of mitochondria and ER;

DistanceMap: Distance on the mitochondria between edge of mitochondria and ER;

Canny: Canny Operator to extract the edge of each object;

ConnectedComponents: find connected regions of each mask;

ncon ← 0
initialize MinDistance ← threshold + 1
initialize DistanceMap ← threshold + 1
overlap ← Mito&ER
edgemito ← Canny(Mito)
edgeer ← Canny(ER)
nmito ← ConnectedComponents(Mito)
for Every pixel (i,j) of edgeer do
if not overlap[i][j] then
for threshold pixels (x,y) of mitochondria around ER do
if not overlap[x][y] then
Minimum distance der between Mitochondria and ER
MinDistance[i][j] ← der
end if
end for
end if
end for
for Every pixel (i,j) of edgemito do
if not overlap[i][j] then
flagcontact ← False
for theshold pixels (x,y) of ER around mitochondria do
Minimum distance dmito between Mitochondria and ER
if dmito < MinDistance[x][y] + 1 then
DistanceMap[i][j] ← dmito
flagcontact ← True
end if
end for
if flagcontact then
ncon ← ncon + 1
end if
end if
end for
if overlap and edgemito then
DistanceMap[i][j] ← 0
end if
lenmito ← SUM(edgemito)
lencon ← SUM(DistanceMap[i][j] < threshold + 1)

## Computation settings

For the Mask R-CNN baseline, we directly used its public implementation (https://github.com/matterport/Mask_RCNN). For the U-Net baseline, we used the PyTorch module (https://github.com/qubvel/segmentation_models.pytorch/tree/V0.1.0). For contact analysis, we implemented our algorithms based on Python. Computational resources were common settings for lab workstations. The operating system was Ubuntu 14.04 LTS 64-bit. We trained and inferenced models on a single GeForce GTX 1080Ti graphics processing unit. The other hardware

information is as follows: 128G memory and 40 core Intel Xeon CPU E5-2640 v4 @ 2.40 GHz.

## Quantification and statistical analysis

The quantitative parameters generated from DeepContact were the number of mitochondria (Mito_number), total perimeter (p) of the mitochondria (Mito_length), mean perimeter of the mitochondria (Mito_length_mean), total number of ER-Mito MCSs (ER-Mito_contact_number), total perimeter and total area of the ER, factor of mitochondrial and ER elongation (p2/[4π × area]; Zhao et al., 2017), and total length of the ER-Mito MCS (ER-Mito_contact_length). Significance was calculated by two-tailed unpaired $t$ test with Welch's correction (nonparametric, do not assume equal SDs). Data distribution was assumed to be normal but was not formally tested. All types of plots, heatmaps, and statistics were generated using GraphPad Prism 8.3.0. Individual values in each plot indicate a summed or averaged value of the corresponding parameters from one micrograph, representing the organelle information from the main body part of a cell. The sample size in each experimental setting was 30 unless otherwise indicated. Measurements were from distinct samples.

## AMIRA version of DeepContact

To develop a user-friendly GUI, we incorporated DeepContact into AMIRA (Stalling et al., 2005), a popular commercial software for electron microscope image analysis. We also provide a detailed tutorial to guide users in installing and using the DeepContact module on AMIRA. The tutorial is available on GitHub: https://github.com/LX-doctorAI1/DeepContact/blob/main/DeepContact_Tutorial.pdf.

## Online supplemental material

Fig. S1 shows the workflow in the incorporation of top likelihood sampling and similarity loss into the Mask R-CNN framework and its effectiveness by virtualization. Fig. S2 shows the conformities of organelle segmentation by DeepContact compared with manual annotation and ER staining. Fig. S3. shows the abnormalities of mitochondria in artificial tether or contact indicator expression systems. Fig. S4 shows the profiling of MCS and organelle morphology by DeepContact in various experimental systems. Fig. S5 shows the workflow for analyzing ER-Mito contact in Sertoli cells using DeepContact. Table S1 is the conformities of DeepContact organelle models with manual annotation. Table S2 is the comparison of ER segmentation using different models. Tables S3, S4, and S5 are the time consumption comparison between manual annotation and DeepContact analysis of the ER-Mito MCS and LD-Mito MCS of a cultured cell, and of the ER-Mito MCS of a Sertoli cell in seminiferous epithelial tissue.

## Data availability

Several representative trained models (https://doi.org/10.6084/m9.figshare.19845940), corresponding training data annotated by Labelme, and example images for testing (https://doi.org/10.6084/m9.figshare.19898404.v3) are publicly available at the figshare repository. The DeepContact source code is available on GitHub: https://github.com/LX-doctorAI1/DeepContact.

## Acknowledgments

We are grateful to Xueke Tan, Zhongshuang Lv, and Can Peng for helping with sample preparation; Jianguo Zhang, Xing Jia, Tongxin Niu, Yun Feng, and Fei Sun for supporting scanning EM imaging and software incorporation at the Center for Biological Imaging (CBI), Institute of Biophysics, Chinese Academy of Sciences. We thank Wanzhong He for supporting tissue sample preparation; YuFan Luo for setting up Labelme software; Junjie Hou, Feng Han, and Yanhong Xue for critical suggestions in the statistical analysis; Ye Jia for preparation of the construct of rapamycin-induced ER-Mito tethering indicator; Chao Tong for providing Split-GFP indicator cell clone; Maoge Zhou for helping with culture and biochemical experiments; Rex A. Hess for precious time and kind instructions on staging of the seminiferous epithelial EM samples; Hongyu Zhao, Yan Zhao, and Hong Zhang for providing data for VMP1 KO, VAPa/b KD, and ATL KO EM data; Yinhao Li for application of the labeling tools; Dee Li and Dong Guo for application of the DeepContact software; Sitong Liu for figure editing; and Jingze Lu and Yuchen Yao for management of the experimental systems.

T. Xu is supported by National Natural Science Foundation of China (grant no. 32027901 and 31730054) and National Key Research and Development Program of China (grant no. 2017YFA0504702). J. Hu is supported by the National Natural Science Foundation of China (grant no. 91854202) and the Strategic Priority Research Program of the Chinese Academy of Sciences (XDB39000000). L. Xiao is supported by the National Natural Science Foundation of China (grant no. 31900979) and CCF-Tencent Open Fund. L. Liu is supported by National Natural Science Foundation of China (grant no. 31700748).

The authors declare no competing financial interests.

Author contributions: Conceptualization: J. Hu, T. Xu, L. Xiao, L. Liu. Methodology: L. Xiao, L. Liu, S. Yang, Y. Liu, X. Li. Investigation: L. Liu, J. Hu, L. Xiao, S. Yang. System Design and Programming: L. Xiao, S. Yang, Y. Liu. Visualization: L. Liu, L. Xiao, S. Yang. Supervision: L. Xiao, T. Xu, J. Hu. Writing—original draft: L. Liu, L. Xiao, S. Yang. Writing—review & editing: J. Hu, T. Xu, L. Liu, L. Xiao.

Submitted: 30 June 2021

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

# Supplemental material

Figure S1. **Effectiveness of the incorporation of top likelihood sampling and similarity loss into the Mask R-CNN framework. (a)** The network adopts ResNet101-FPN as the backbone. A top likelihood loss is applied to sample the top scoring negative anchors when training RPN and the top scoring negative proposals when training the following branches. A similarity loss is simultaneously added on the fc-layer of the Fast R-CNN branch to discriminate negative proposals from positive ones. **(b)** Original image. **(c)** Image with ground-truth labels. **(d)** Prediction from baseline Mask R-CNN. **(e)** Prediction of Mask R-CNN with top likelihood sampling and similarity loss added. Red dotted rectangles indicate a reduction of false positives.

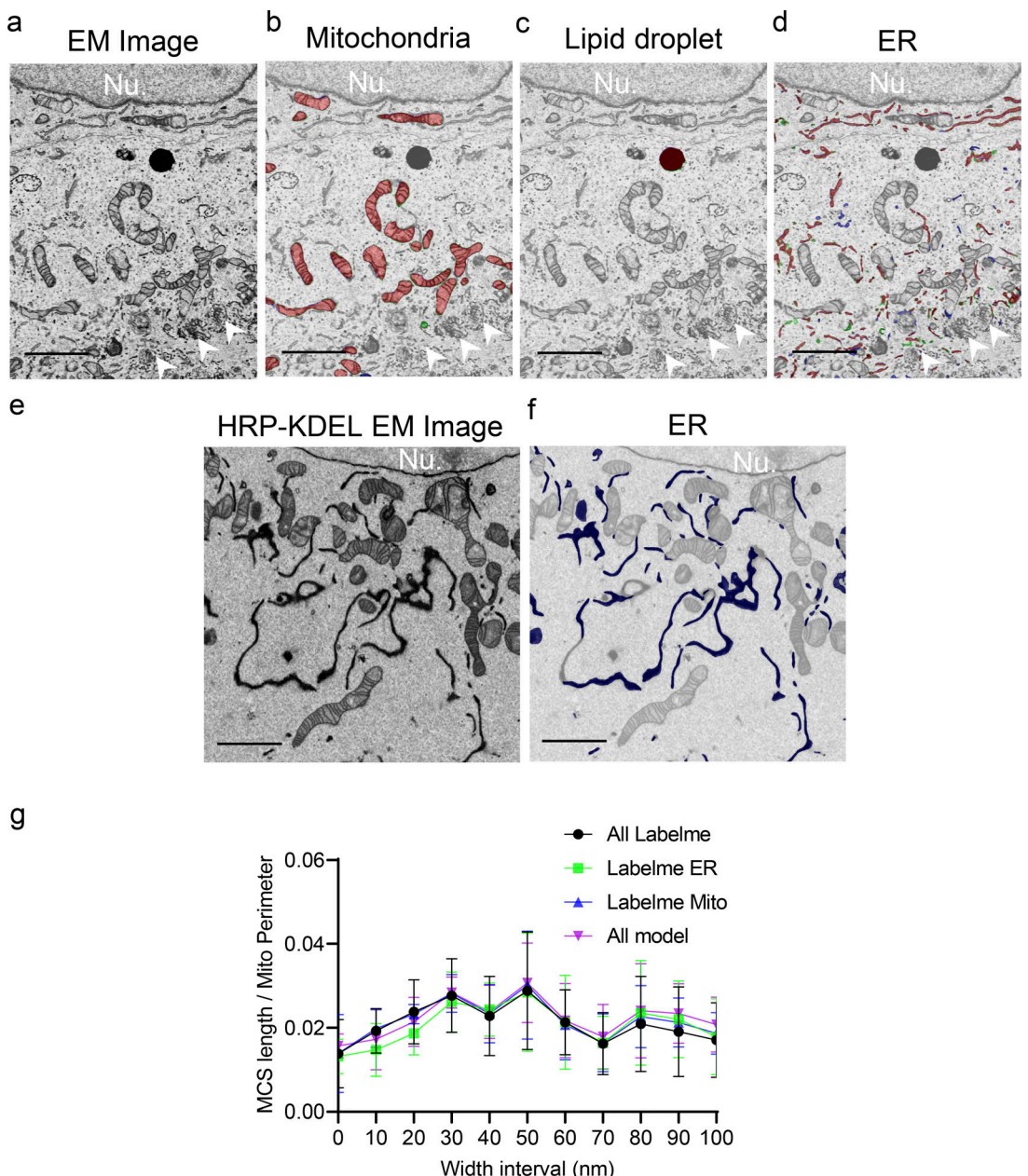

Figure S2. **Conformities of organelle segmentation by DeepContact with manual and ER staining. (a)** Original patch image of a U-2 OS cell. **(b–d)** Segmentation of mitochondria (b), LD (c), and ER (d) by manual annotation and DeepContact. Red-colored areas, intersections of DeepContact segmentation with manual annotation; green, manual annotation only; and blue, DeepContact segmentation only; Nu, nucleus. Scale bar, 2 μm. Arrowheads indicate Golgi apparatus. **(e)** Original patch EM image of a HRP-KDEL–transfected U-2 OS cell stained with DAB. **(f)** Segmentation of ER by DeepContact. Scale bar, 2 μm. **(g)** Comparison of MCS length/mitochondria perimeter ratio in 0- to 100-nm-width intervals with the combined boundary lines of ER and mitochondria segmented by the Labelme or DeepContact models. The sample size is six. Data are presented as mean values with 95% confidence intervals.

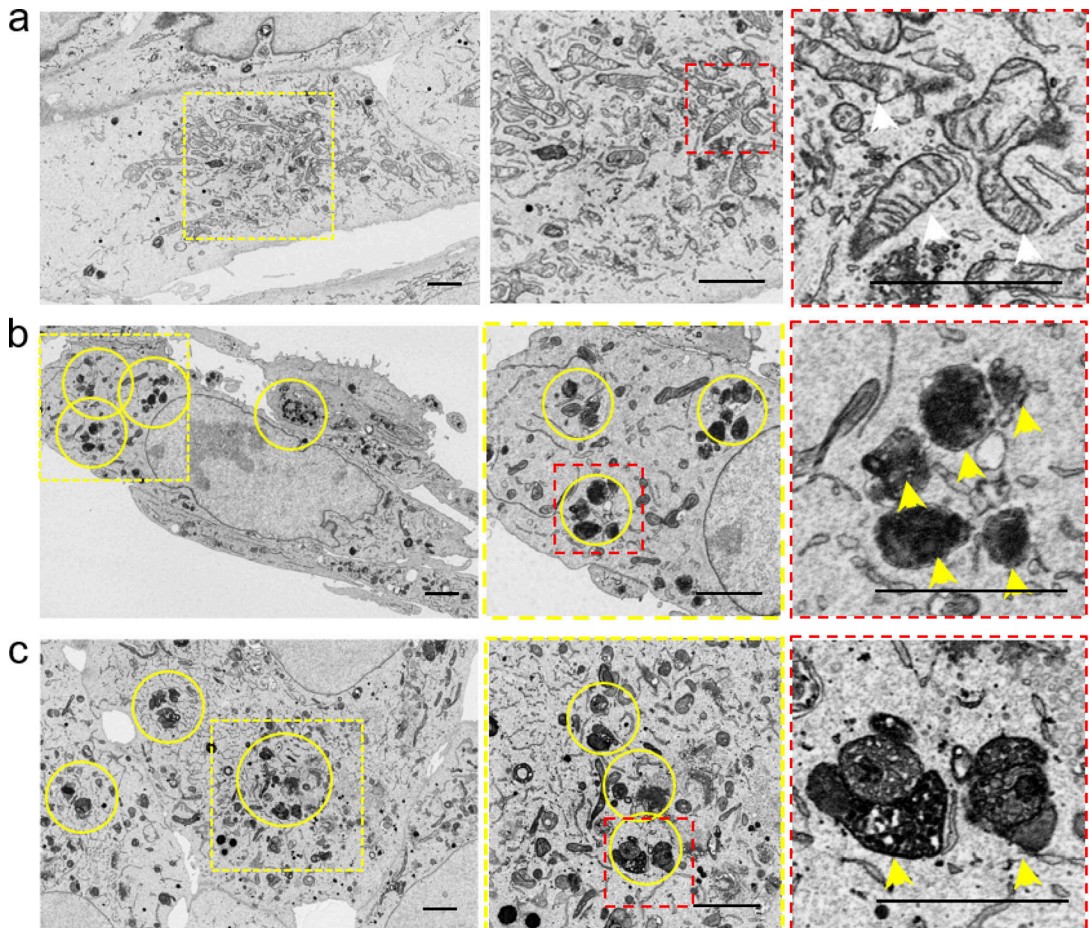

Figure S3. **Mitochondrial defects in artificial tether or contact indicator expression systems. (a)** Normal mitochondrial morphologies in wild-type U-2 OS cells. **(b)** Degraded morphologies in a stable cell line reconstructed with the rapamycin-induced ER-Mito tethering system. **(c)** Aggregated morphologies in single-cell clones reconstructed with ER-Mito MCS. Scale bar, 2 μm.

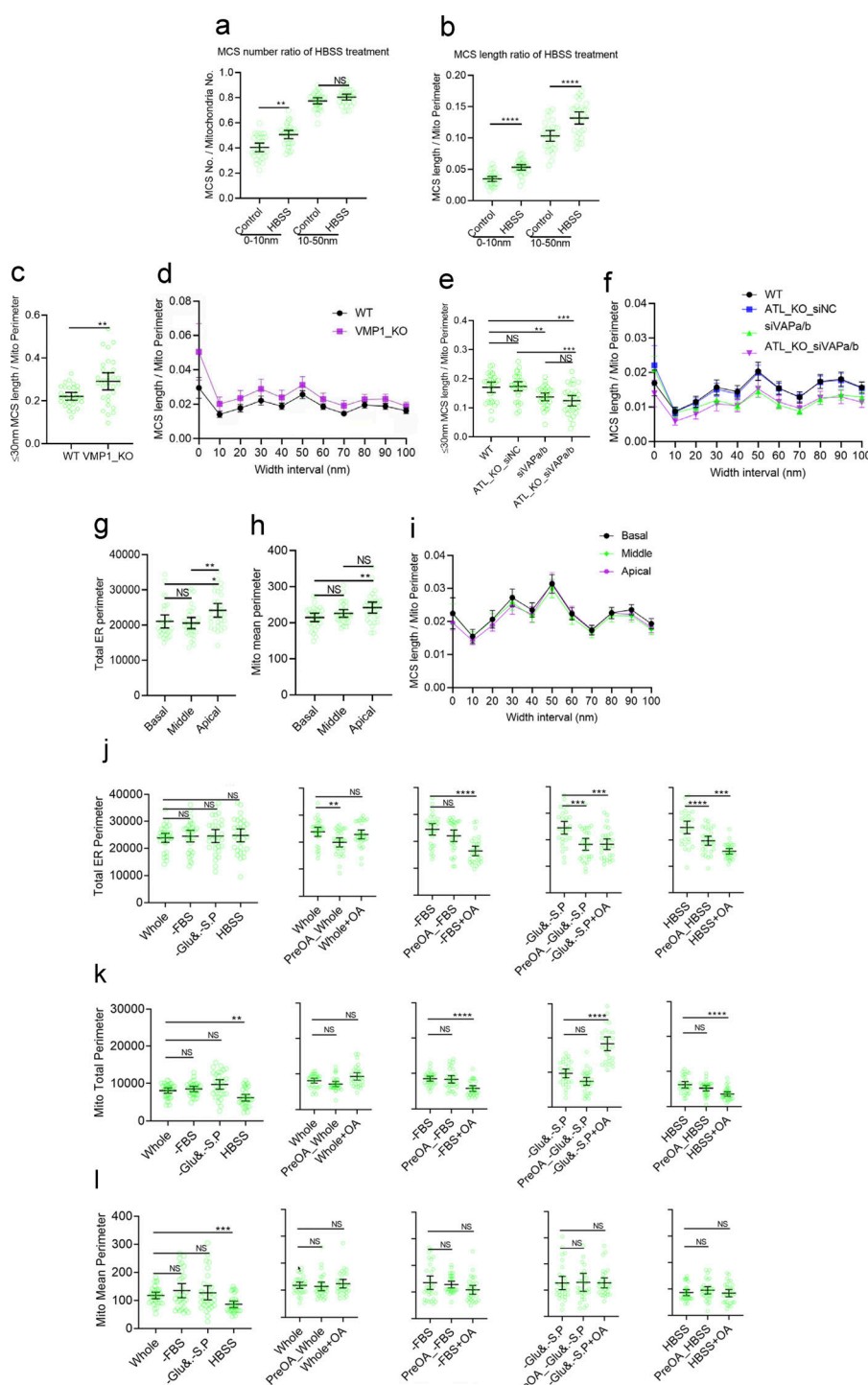

Figure S4. **Profiling of MCS and organelle morphology by DeepContact in various experimental systems. (a)** Alterations in the MCS number relative to the total mitochondria number in width intervals of 0–10 or 10–50 nm with HBSS starvation. **(b)** Alterations in MCS length relative to the mitochondrial perimeter. **(c)** Alterations in the Mito-ER MCS length ratio in 0- to 30-nm-width intervals with VMP1 deletion. **(d)** MCS length ratio profiling in 0- to 100-nm-width intervals with VMP1 deletion. **(e)** Alterations in the MCS length ratio in 0- to 30-nm-width intervals with VAPa/b or ATL deletion and their combined depletion. **(f)** MCS length ratio profiling in 0- to 100-nm-width intervals with VAPa/b or ATL deletion and their combined depletion. **(g)** Total ER perimeter in sections from the basal, middle, and apical level of the adherent U-2 OS cells. **(h)** Mean mitochondrial perimeter in sections from the basal, middle, and apical level of the adherent U-2 OS cells. **(i)** MCS length ratio in MCS 0- to 100-nm-width intervals in sections from the basal, middle, and apical level of the adherent U-2 OS cells. **(j)** Total perimeter of ER in complete medium (Whole), FBS starvation (−FBS), glucose and sodium pyruvate starvation (−Glu&. −S.P.), and HBSS starvation conditions in combination with OA treatment 10 h before (Pre-OA) or during (OA) starvation treatment of the cells. **(k)** Total perimeter of mitochondria in Whole, −FBS, −Glu&. −S.P., and HBSS conditions combined with Pre-OA or OA treatments. **(l)** Mean perimeter of mitochondria in Whole, −FBS, −Glu&. −S.P., and HBSS conditions combined with Pre-OA or OA treatments. The sample size of each experimental setting is 30, and individual dots in the plot represent the mean value of a 15 × 10–μm cellular image. Bars indicate 95% confidence intervals.

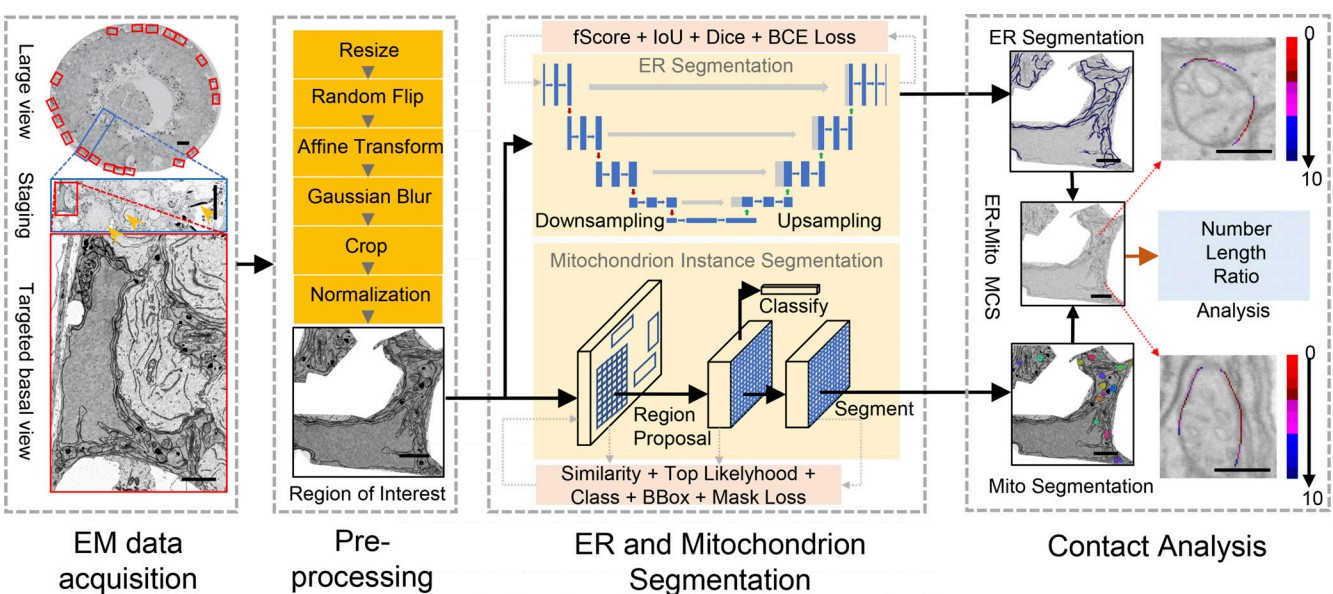

Figure S5. **The workflow for analyzing ER-Mito contact in Sertoli cells using DeepContact.** The large image was acquired from EM, and the staging image was cropped from the large view. The annotations include the plasma membrane, mitochondrion, and ER of the Sertoli cell. The plasma membrane is used to extract the RoI, which is the area within the Sertoli cell territory. We preprocess the staging image in a series of operations, including resizing, random flip, affine transform, Gaussian blur, cropping, and normalization, to obtain patches with a standard size of 1,024 × 1,024 pixels and resolution of 10 nm. The IoU, which represents the ratio of the RoI area according to the whole image, is computed for each sample. Only samples with IoU > 0.15 are selected for analysis. We adopted fScore loss, IoU loss, Dice loss, and BCE loss to train the ER segmentation model and similarity loss, top likelihood loss, CE loss, bounding box regression loss, and mask IoU loss to train the mitochondria segmentation model. The output of the ER segmentation model is the whole ER region. The output of the mitochondria segmentation model is the instance of an individual mitochondrion. The number, perimeter, area, ratio, and MCS profile are calculated by DeepContact using ER areas and mitochondrial instances. The MCS profile includes the length and ratio for 1- to 10-pixel-width intervals, represented by different colors. Scale bar in large and middle size view of the left panel, 10 μm; in targeted basal view of the left panel and patch view, 2 μm; in the enlarged view of the right panel, 0.5 μm.

**Provided online are Tables S1, S2, S3, S4, and S5. Table S1 shows performance in ER segmentation using different models. Table S2 shows conformities of DeepContact organelle models with manual annotation. Table S3 shows time consumption comparison between Labelme manual annotation and DeepContact analysis of the ER-Mito MCS of a cultured cell. Table S4 shows time consumption comparison between Labelme manual annotation and DeepContact analysis of the LD-Mito MCS of a cultured cell. Table S5 shows time consumption comparison between Labelme manual annotation and DeepContact analysis of the ER-Mito MCS of a Sertoli cell in seminiferous epithelial tissue.**

