## [Peer Review File · The Journal of Cell Biology]

DeepContact: High throughput quantification of membrane contact site on electron microscopy imaging

Liqing Liu, Shuxin Yang, Yang Liu, Xixia Li, Junjie Hu, Li Xiao, and Tao Xu

Corresponding Author(s): Li Xiao, Institute Of Computing Technology

Review Timeline:

Submission Date:	2021-06-30
Editorial Decision:	2021-09-27
Revision Received:	2022-03-04
Editorial Decision:	2022-05-09
Revision Received:	2022-05-31
Editorial Decision:	2022-06-27
Revision Received:	2022-07-06

Monitoring Editor: Jodi Nunnari

Scientific Editor: Dan Simon

Transaction Report:

DOI: <https://doi.org/10.1083/jcb.202106190>

September 27, 2021

Re: JCB manuscript #202106190

Prof. Li Xiao
Institute Of Computing Technology
No.6 Zhongguancun Road
Haidian District
Beijing, Beijing 100090
China

Dear Prof. Xiao,

Thank you for submitting your manuscript entitled "DeepContact: High throughput quantification of MCS based on electron microscopy imaging." Thank you for your patience with the peer review process. The manuscript has now been evaluated by expert reviewers, whose reports are appended below. Unfortunately, after an assessment of the reviewer feedback, our editorial decision is against publication in JCB.

You will see that the reviewers feel that DeepContact could be a potentially valuable tool but express significant concerns regarding the methodology and validation of the algorithm and thus are not convinced that it accurately measures membrane contact sites.

Although your manuscript is intriguing, we feel that the points raised by the reviewers are more substantial than can be addressed in a typical revision period. If you wish to expedite publication of the current data, it may be best to pursue publication at another journal. Our journal office will transfer your reviewer comments to another journal upon request.

Given interest in the topic, we would be open to a re-evaluation of a substantially revised version of the study, but we believe that this would entail a significant amount of additional experimental work. If you would be interested in this possibility, we request that you submit a revision plan that includes a point-by-point response to each of the reviewer comments, and how you would address them. We will ask both reviewers for their feedback on whether this revision plan fully addresses their concerns. If we feel it is necessary we may also recruit a third reviewer for the revised manuscript.

Regardless of how you choose to proceed, we hope that the comments below will prove constructive as your work progresses. We would be happy to discuss the reviewer comments further once you've had a chance to consider the points raised in this letter. You can contact the journal office with any questions, cellbio@rockefeller.edu or call (212) 327-8588.

Thank you for thinking of JCB as an appropriate place to publish your work.

Sincerely,

Jodi Nunnari, Ph.D.
Editor-in-Chief
The Journal of Cell Biology

Dan Simon, Ph.D.
Scientific Editor
The Journal of Cell Biology

Reviewer #1 (Comments to the Authors (Required)):

The manuscript by Liu et al introduces a deep learning-based method for segmentation of organelles and analysis of their contact sites in label-free 2D EM images. I am not an expert on the biology of organelle contact and will leave it to other reviewers to judge the significance of detected changes, limiting this review to algorithmic aspects. The manuscript addresses an important problem, organelle segmentation is a difficult and error prone task that many biologists would like to see automated. I have the following questions/comments for the authors:

- My biggest concern lies with the quantitative evaluation of the method. A human expert has annotated 10 images and the test set, the basis for Suppl Table 2, is cut out of the same images. This is not an unbiased analysis, it does not necessarily provide a good estimate of accuracy on images completely not seen in training. Would it be possible to have a full leave-one-image-out

cross-validation done?

- Continuing on this point, how big are the 10 images, in pixels? How many organelles were segmented there? How many contact sites, if any?
- Why is an image-level metric used for evaluation instead of an object-based metric? Contacts happen between objects and it would be important to know how well each of them is segmented.
- I must be missing something, but I don't see an evaluation on the accuracy of the morphometric estimates compared to manual estimates. My worry here is that the automatic method might have a bias which does not show in the object segmentation evaluation, but would affect the contact site measurements. For example, if the algorithm consistently segments objects as a little shrunk or a little dilated, the authors' segmentation metric would still look good, but the contact site measurements might be off.

Other major points:

- The methods used by authors have been introduced before, so the main contribution appears to be their application to an important and challenging problem. While I fully support publication of new applications of known methods, here it is not clear to me how the readers are supposed to benefit from DeepContact? Where and when will the software be made available? Is it open source and if not, why not? Does it have documentation, binaries, user contact possibilities? Where and when will the reference data become available?
- from the Discussion, it appears that the algorithm needs to work on the special ROTO staining, but this is not mentioned in the Introduction and only introduced in the middle of the results. For readers who can't see the sample preparation just from the figures, could you clarify this question? Is this staining necessary? How much of a performance gain does it bring?
- How does the active learning strategy incorporate the difficult cases not seen in training? In other words, how does the exploration phase happen if the expert is mostly looking for unannotated images similar to the ones in the training set?
- What do the arrows in Fig. 4 show?
- Suppl fig 1 - what are the red numbers, they are not readable? Can they become a table or a summary score of the segmentation improvement?

Reviewer #2 (Comments to the Authors (Required)):

Liu et al. have developed « DeepContact » a deep learning protocol conceived to quantify membrane contact sites (MCS) by EM, that optimizes organelle segmentation and contact analysis. Deep learning methods have been already developed to analyze MCS in 3D EM. However, they require high performance computing resources and long processing time for the analysis of a limited number of cells.

The authors designed "DeepContact" to analyze MCS in label-free 2D EM. As compared to the classical manual annotation-based organelle segmentation, "DeepContact" has the advantage to automatize the procedures of processing the 2D EM slices and provide an unbiased and efficient way to quantify MCS in a larger number of cells.

In this study, the accuracy of DeepContact in organelles segmentation was validated by comparison to a manual method in randomly acquired images. Deep Contact was also used to profile new rearrangements of MCS upon different nutrient conditions and changes in ER-mitochondria MCS in the Sertoli cells during the seminiferous epithelial cycle.

However, still some key questions remains to be addressed to confirm "DeepContact" as a powerful tool to analyze MCS :

- The authors could not quantify ER-mitochondria contacts upon expression of a ER-mito tethering system as this system caused frequent mitochondrial abnormalities. There are other ways (more physiological and based on published works) to modulate ER-mitochondria tethering by expressing or downregulating tethering proteins. For instance, overexpression of the ER-mitochondria tether PTP51 has been shown to increase the ER-mitochondria contacts and its knockdown instead decreases ER-mitochondria MCS (of about 30-40%) (PMID: 24893131, PMID: 27113756, etc..). Alternatively, PDZD8 has also been shown to decrease ER-mitochondria contacts when downregulated (PMID: 29097544). It would be important to know if DeepContact can allow to detect these changes, that are often not so strong as those induced by an artificial system, but have more physiological relevance. Testing this will be important to confirm the power of DeepContact to study the effect of membrane tethers on MCS.

- Can "DeepContact" be used in 2D EM sections where the lumen of organelles (i.e. ER or Golgi) are stained (for example by HRP-based methods)? Staining of these organelles is important for accurate quantification of contact sites. Indeed, conventional EM does not allow to accurately discriminate between organelles such as ER or Golgi (for example when the ER is cut transversally seems a small vesicle) and requires a perfect the preservation of sample morphology. Importantly, Golgi can also

form contacts with the ER and the mitochondria (i.e. see J. Prudent's work PMID: 32193326). Thus, HRP-KDEL (or other similar methods) staining of the ER is often used with success to analyse MCS involving the ER, including ER-mitochondria MCS. It would be important that the authors compare the accuracy of Deep Contact in analyzing ER-mitochondria MCS on unstained vs HRP-stained EM slices (reporting as % the length of mitochondria surface in contact with ER).

Minor points:

- Page 8 line 189= The authors says that "cultured cells are usually trypsinized and centrifuged prior to conventional EM processing". Thus, they optimized a saffire disk-assisted sample preparation of adherent cells to maintain the native topology of the intracellular organization.

The first sentence should be removed or rephrased, as most of conventional EM protocols are done on monolayers of cells adherent on coverslips (i.e. PMID: 33772156, PMID: 23791178) or by scraping cells already fixed on plastic dishes, that preserve the shape of the cell and the organization of the intracellular organelles.

The authors say that have optimized adherent cell EM preparation procedure. The fixation-embedding protocol used here seems to me quite classical.

- Also, what is the advantage of using here saffire disk versus glass coverslips? To my knowledge saffire disks are just more expensive and complicate to manipulate given their very small size. They are usually used for high-pressure freezing/freeze substitution, a method that allow to preserve the organelles membrane in a more native state. But this was not the method used to prepare the samples in this study.

Reviewer #1

The manuscript by Liu et al introduces a deep learning-based method for segmentation of organelles and analysis of their contact sites in label-free 2D EM images. I am not an expert on the biology of organelle contact and will leave it to other reviewers to judge the significance of detected changes, limiting this review to algorithmic aspects. The manuscript addresses an important problem, organelle segmentation is a difficult and error prone task that many biologists would like to see automated. I have the following questions/comments for the authors:

- *My biggest concern lies with the quantitative evaluation of the method. A human expert has annotated 10 images and the test set, the basis for Suppl Table 2, is cut out of the same images. This is not an unbiased analysis, it does not necessarily provide a good estimate of accuracy on images completely not seen in training. Would it be possible to have a full leave-one-image-out cross-validation done?*
- *, 6 on this point, how big are the 10 images, in pixels? How many organelles were segmented there? How many contact sites, if any?*

We thank the reviewer for pointing this out. The images used previously are 2048 x 2048 in pixels. The numbers of segmented organelles are 397.1-Mito, 360.7-ER and 102.2-LD by manual annotation, and 387.8-Mito, 423.8-ER and 101.6-LD by DeepContact. Contact sites of ER-Mito and LD-Mito are 187 and 19.2, respectively. For unbiased analysis, we redo the comparison using 6 images unseen by the program, and perform the validation as suggested. Images used currently are 2048 x 2048 in pixels. The numbers of segmented organelles are 252-Mito, 210-ER and 43-LD by manual annotation, and 252-Mito, 210-ER and 44-LD by DeepContact. Contact sites of ER-Mito and LD-Mito are 220 and 20 respectively. We remade the table to show conformities of DeepContact organelle models with manual annotation. (Supplementary Table 2) Related information were updated in the manuscript.

	Mito_match/Mito_manual	ER_match/ER_manual	LD_match/LD_manual
Mean±SD	97.63±3.26%	87.71±5.87%	98.48±3.71%

*Mito, mitochondria; ER, endoplasmic reticulum; LD, lipid droplet. n = 6, values were represented as mean ± SD.

- *Why is an image-level metric used for evaluation instead of an object-based metric? Contacts happen between objects and it would be important to know how well each of them is segmented.*

We segment the ER with semantic segmentation method (image-level metric), and mitochondria (Mito) and lipid droplet (LD) with instant segmentation method (object-based metric). Both Mito and LD are composed by physiologically and functionally individualized entities. It is proper to use object-based metric, such as Mask R-CNN, to segment these organelles in individualized objects. By contrast, the ER is an interconnected and contiguous network, which mostly represents as cisterna-like structures in EM images. To segment the ER at image level may help to model the integrative patterns of ER, and to exclude other cisterna-like organelle networks from the model, including the Golgi apparatus (See figure below). It would be very difficult

to discriminate ER cisterna from Golgi cisterna or other similar organelles in EM data, if segment all cisterna at a single object level. We have added additional data (**Supplementary Figure 2a, c**) and the explanation in the text accordingly.

Golgi cisterna are excluded from ER segmentation by DeepContact. a and b, original view of EM data. a' and b' ER segmentation by DeepContact. Arrows indicate Golgi apparatus.

- I must be missing something, but I don't see an evaluation on the accuracy of the morphometric estimates compared to manual estimates. My worry here is that the automatic method might have a bias, which does not show in the object segmentation evaluation, but would affect the contact site measurements. For example, if the algorithm consistently segments objects as a little shrunk or a little dilated, the authors' segmentation metric would still look good, but the contact site measurements might be off.

We thank the reviewer for the suggestion. To address this concern, we employed the 6 images newly annotated by “Labelme” software. Ratio of MCS length/Mito perimeter in 0-100 nm width intervals was compared with the combined boundary lines of ER and Mito segmented by either Labelme or DeepContact. MCS ratios in all width intervals ranges appeared comparable between each combination, indicating accurate boundary detection of the organelles by DeepContact (**Supplementary Figure 2g**).

Notably, we profile contacts in 10 nm intervals and compare contact systematically throughout the 0-100 nm spectra, which in principle would minimize errors caused by the morphometric estimates. In addition, qualitative conclusion between same types of samples would not be influenced by minor system errors if there are any.

Comparison of MCS length/Mito perimeter ratio in 0-100 nm width intervals with the combined boundary lines of ER and Mito segmented by Labelme or DeepContact models. The sample size is 6. Mean values and bars are indicated as mean with 95% CIs.

- The methods used by authors have been introduced before, so the main contribution appears to be their application to an important and challenging problem. While I fully support publication of new applications of known methods, here it is not clear to me how the readers are supposed to benefit from DeepContact? Where and when will the software be made available? Is it open source and if not, why not? Does it have documentation, binaries, user contact possibilities? Where and when will the reference data become available?

We fully agree that the methods should be made available to all readers. We have previously uploaded the software on GitHub (link: <https://github.com/LX-doctorAI/DeepContact>). We have added this point in the text. We plan to provide additional documents in the future to guide user through the process.

- from the Discussion, it appears that the algorithm needs to work on the special ROTO staining, but this is not mentioned in the Introduction and only introduced in the middle of the results. For readers who can't see the sample preparation just from the figures, could you clarify this question? Is this staining necessary? How much of a performance gain does it bring?

We apologize for causing confusion here. The ROTO staining was introduced to enhance the contrast of the lipid-based membrane boundaries of the organelles for a more accurate boundary detection by DeepContact. However, it is not required for the methods. We have modified the text accordingly to clarify the point.

- How does the active learning strategy incorporate the difficult cases not seen in training? In other words, how does the exploration phase happen if the expert is mostly looking for unannotated images similar to the ones in the training set?

The active learning strategy is applicable when there are organelles on the testing set which are missed in prediction after each round of training. We manually check these difficult cases and ensure that additional training set (unannotated images) contains sufficient cases with similar features for the model to be improved. The procedure is applied recursively until the model can predict all the organelles on the testing set. We have adjusted the wording in the text to clarify this point.

- *What do the arrows in Fig. 4 show?*

Arrowheads indicate mitochondria that are not segmented initially, but by refined model through active learning processes. We have clarified this in the legends.

- *Suppl fig 1 - what are the red numbers, they are not readable? Can they become a table or a summary score of the segmentation improvement?*

Red numbers in Supplementary Fig. 1d were probability scores of predicted mitochondria, which were not necessary in assisting the result demonstration. We have re-run the testing processes and removed the numbers.

Reviewer #2

Liu et al. have developed “DeepContact” a deep learning protocol conceived to quantify membrane contact sites (MCS) by EM, that optimizes organelle segmentation and contact analysis. Deep learning methods have been already developed to analyze MCS in 3D EM. However, they require high performance computing resources and long processing time for the analysis of a limited number of cells.

The authors designed “DeepContact” to analyze MCS in label-free 2D EM. As compared to the classical manual annotation-based organelle segmentation, “DeepContact” has the advantage to automatize the procedures of processing the 2D EM slices and provide an unbiased and efficient way to quantify MCS in a larger number of cells.

In this study, the accuracy of DeepContact in organelles segmentation was validated by comparison to a manual method in randomly acquired images. Deep Contact was also used to profile new rearrangements of MCS upon different nutrient conditions and changes in ER-mitochondria MCS in the Sertoli cells during the seminiferous epithelial cycle.

However, still some key questions remain to be addressed to confirm “DeepContact” as a powerful tool to analyze MCS:

- The authors could not quantify ER-mitochondria contacts upon expression of a ER-mito tethering system as this system caused frequent mitochondrial abnormalities. There are other ways (more physiological and based on published works) to modulate ER-mitochondria tethering by expressing or downregulating tethering proteins. For instance, overexpression of the ER-mitochondria tether PTPIP51 has been shown to increase the ER-mitochondria contacts and its knockdown instead decreases ER-mitochondria MCS (of about 30-40%) (PMID: 24893131, PMID: 27113756, etc..). Alternatively, PDZD8 has also been shown to decrease ER-mitochondria contacts when downregulated (PMID: 29097544). It would be important to know if DeepContact can allow to detect these changes, that are often not so strong as those induced by an artificial system, but have more physiological relevance. Testing this will be important to confirm the power of DeepContact to study the effect of membrane tethers on MCS.

We thank the reviewer for the valuable suggestion. To address this concern, we used deletion cell lines that are known to alter ER-Mito contact. Autophagic protein EPG-3/VMP1 has been shown to regulates contact of the ER with numerous organelles, including mitochondria (PMID 28890335). When deleted in cells, the ER-mito contact is increased. By contrast, contact site adapter protein VAPA/B is well known to mediate a variety of ER-based contact. In the case of ER-mito contact, PTPIP51 (as suggested by the reviewer) or Vps13D bridges the two organelles in a VAP-dependent manner (PMID 33891013). We reason that depletion of VAPA/B would reduce ER-mito contact. As a control, we used ATL-depleted cells. ATL is ER membrane fusogen, the lack of which causes an aberrant ER morphology but with no known impact on Mito-ER contact. DeepContact obtained consistent results, with improved accuracy, when compared to the reported functional scenario of these molecular machineries, and previous results obtained by manual segmentation. We have added these new data in the manuscript (**Supplementary Figure 5**).

a, Alterations in the Mito-ER MCS length ratio in 0-30 nm width intervals with VMP1 depletion. b, MCS length ratio profiling in 0-100 nm width intervals with VMP1 depletion. c, Alterations in the MCS length ratio in 0-30 nm width intervals with VAPa/b or ATL-depletion, and their combined depletions respectively. d, MCS length ratio profiling in 0-100 nm width intervals with VAPa/b or ATL-depletion respectively, and their combined depletions. The sample size of each experimental setting is 30, and individual dots in the plot represent the mean value of a $15\ \mu\text{m} \times 10\ \mu\text{m}$ cellular image. Mean values and bars are indicated as mean with 95% CIs on a-d.

- Can “DeepContact” be used in 2D EM sections where the lumen of organelles (i.e. ER or Golgi) are stained (for example by HRP-based methods). Staining of these organelles is important for accurate quantification of contact sites. Indeed, conventional EM does not allow to accurately discriminate between organelles such as ER or Golgi (for example when the ER is cut transversally seems a small vesicle) and requires a perfect the preservation of sample morphology. Importantly, Golgi can also form contacts with the ER and the mitochondria (i.e. see J. Prudent's work PMID: 32193326). Thus, HRP-KDEL (or other similar methods) staining of the ER is often used with success to analyze MCS involving the ER, including ER-mitochondria MCS. It would be important that the authors compare the accuracy of Deep Contact in analyzing ER-mitochondria MCS on unstained vs HRP-stained EM slices (reporting as % the length of mitochondria surface in contact with ER).

As suggested, we implanted the HRP-based methods to verify the accuracy of the ER segmentation. We prepared parallel samples of cells transfected with either empty

vector or HRP-KDEL. In addition, HRP-KDEL transfected cells was duplicated for EM imaging, one without staining and the other stained. Such three samples were subjected to DeepContact comparison. Because the relative contrast of ER and mitochondria are altered in stained HRP-KDEL transfected cells, we refined DeepContact models by additional trainings using active learning strategy. The patterns of ER networks in three kinds of preparations appeared similar, with DAB stained ER networks match well with the DeepContact segmentation (**Supplementary Figure 2e**). MCS Ratio in 0-100 nm width intervals were also comparable in all three preparations (**Supplementary Figure 2f**).

a, Segmentation of Mito, ER and MCS in cells transfected with empty vector without staining (Vesicle No stain), and HRP-KDEL transfected cells without (HRP-KDEL No stain) or with (HRP-KDEL DAB stain). b, MCS length ratio profiling in 0-100 nm width intervals in cells transfected with empty vector without staining (Vesicle No stain), and HRP-KDEL transfected cells without (HRP-KDEL No stain) or with (HRP-KDEL DAB stain). The sample size of each experimental setting is 30. Mean values and bars are indicated as mean with 95% CIs.

Minor points

- Page 8 line 189= The authors say that "cultured cells are usually trypsinized and centrifuged prior to conventional EM processing". Thus, they optimized a sapphire disk-assisted sample preparation of adherent cells to maintain the native topology of the intracellular organization.

The first sentence should be removed or rephrased, as most of conventional EM protocols are done on monolayers of cells adherent on coverslips (i.e. PMID: 33772156, PMID: 23791178) or by scraping cells already fixed on plastic dishes, that preserve the shape of the cell and the organization of the intracellular organelles. The authors say that have optimized adherent cell EM preparation procedure. The fixation-embedding protocol used here seems to me quite classical.

We apologize for the confusion here. We have deleted the sentence accordingly.

- Also, what is the advantage of using here sapphire disk versus glass coverslips? To my knowledge sapphire disks are just more expensive and complicate to manipulate given their very small size. They are usually used for high-pressure freezing/freeze.

The advantage of the sapphire disk is that it is the same size as the EM grid, which not only minimizes the cropping of the embedded materials, but also reduces usage of toxic fixatives. We used domestically produced sapphire disk, making it an economic choice. We have tuned down the part for choosing sapphire disk accordingly.

May 9, 2022

Re: JCB manuscript #202106190R-A

Prof. Li Xiao
Institute Of Computing Technology
No.6 Zhongguancun Road
Haidian District
Beijing, Beijing 100090
China

Dear Prof. Xiao,

Thank you for submitting your revised manuscript "DeepContact: High throughput quantification of membrane contact site based on electron microscopy imaging." Thank you for your patience with the peer review process. The manuscript was seen by original Reviewer #2 but unfortunately Reviewer #1 was not available. We therefore recruited Reviewer #3 who was specifically asked to assess whether the issues raised by Reviewer #1 were addressed.

Although Reviewer #2 now recommends acceptance, you will see that Reviewer #3 feels that you have not adequately addressed points that were raised in the initial review, which consequently preclude publication of the current version of the manuscript. However, we feel that the work has substantially improved and could be suitable for JCB with additional revisions.

A crucial concern is the evaluation of the automated segmentation algorithm. Reviewer #3 notes that the origin of the 6 new images unseen by the program is not specified and so the possibility of bias in segmentation cannot be excluded. Both reviewers also point out that the software currently lacks any instructions for users on how to use DeepContact which must be provided. There are several other comments that most likely can be addressed by text revisions.

Our general policy is that papers are considered through only one revision cycle; however, given the interest in this work and since the required changes are not major we are open to one additional short round of revision. Please note that we will need to see clear enthusiasm from Reviewer #3 before we can move toward acceptance. However, we will reject the manuscript if this revision is deemed not suitable.

The typical time frame for a final revision is one month but please let us know if you would need more time. Please also submit a cover letter that includes a point by point response to the reviewer comments.

Thank you for this interesting contribution to Journal of Cell Biology. You can contact me or the scientific editor listed below at the journal office with any questions, cellbio@rockefeller.edu or call (212) 327-8588.

Sincerely,

Jodi Nunnari, PhD
Editor-in-Chief
Journal of Cell Biology

Dan Simon, PhD
Scientific Editor
Journal of Cell Biology

Reviewer #2 (Comments to the Authors (Required)):

The authors have sufficiently addressed all my points. Thus I support the publication of their work in Journal of Cell Biology. The authors mention to have uploaded the software for organelle segmentation and contact analysis on GitHub (link: <https://github.com/LX-doctorAI/DeepContact>). It will be indeed important to provide additional documents to guide the users through the process (as they also mention in their answer).

Reviewer #3 (Comments to the Authors (Required)):

Authors have addressed the question on quantitative evaluation of automated segmentation by adding 6 images unseen by the

program and running the validation again. Unfortunately, the origin of these unseen images remain elusive. The images used for analysis come from serially sectioned blocks, and the difference between consecutive sections remains too marginal to exclude the possibility of bias in segmentation. If authors have paid special attention in selecting additional sections, which are not part of the original serial section collection, this is not stated in the manuscript.

Another concern that remain unanswered was the question about how the readers are supposed to benefit from DeepContact. Unlike written in the rebuttal letter, I could not find the link to the software in the revised manuscript. Furthermore, the lack of instructions for the use of the software hinders the testing and using of the software by others. Considering that this manuscript is submitted to the Tools section, it lacks very essential parts such as user-friendly GUI and instructions and tutorials for the use of the software.

All images analyzed in the manuscript are taken with in SEM using special sensitive circular backscatter detector, and it is quite clear that boosted membrane contrast is needed for this type of imaging. High membrane contrast is essential for automated segmentation too. Moreover, it remains unclear whether similar workflow could be applied to thin section TEM images, which is the technique that is much more widely accessible for basic EM users. Authors should be more careful in making statements about the suitability of the method for standard contrast samples, unless they show it.

Finally as a minor comment, I would like to add that "Sapphire disk-assisted ROTO sample preparation procedure" is confusing way of saying that cultured cells were chemically fixed and embedded as adherent monolayers. I still find no good reason for using sapphires instead of glass coverslips. In my experience, embedding as adherent monolayers is nowadays more common way of processing cultured cells than preparing cell pellets. Comparison between the two sample preparation is unnecessary.

Reviewer #2 (Comments to the Authors (Required)):

1. *The authors have sufficiently addressed all my points. Thus I support the publication of their work in Journal of Cell Biology.*

We thank the reviewer for finding our revision sufficient.

2. *The authors mention to have uploaded the software for organelle segmentation and contact analysis on GitHub (link:<https://github.com/LX-doctorAI/DeepContact>). It will be indeed important to provide additional documents to guide the users through the process (as they also mention in their answer).*

We thank the reviewer for the kind suggestion. We have prepared additional documents as suggested. To enable user access of DeepContact, we provide two options by which users can apply the software. The first choice is for users who can employ our models and algorithms directly by programming, which would require basics in computing ([Code available at GitHub: https://github.com/LX-doctorAI1/DeepContact](https://github.com/LX-doctorAI1/DeepContact)). The other is to incorporate DeepContact into Amira, a popular commercial software for EM image analysis (Tutorial available at [GitHub: https://github.com/LX-doctorAI1/DeepContact/blob/main/DeepContact_Tutorial.pdf](https://github.com/LX-doctorAI1/DeepContact/blob/main/DeepContact_Tutorial.pdf)), providing a user-friendly GUI for DeepContact. We have added the information to the *Materials and Methods* section of the manuscript.

Reviewer #3 (Comments to the Authors (Required)):

3. *Authors have addressed the question on quantitative evaluation of automated segmentation by adding 6 images unseen by the program and running the validation again. Unfortunately, the origin of these unseen images remain elusive. The images used for analysis come from serially sectioned blocks, and the difference between consecutive sections remains too marginal to exclude the possibility of bias in segmentation. If authors have paid special attention in*

selecting additional sections, which are not part of the original serial section collection, this is not stated in the manuscript.

We thank the reviewer for reminding us to specify the data origin, which is essential information to demonstrate an unbiased comparison of contact site segmentation between DeepContact and manual estimates. Six images are not part of the serial section collections of any sample blocks ever employed for the training of DeepContact models, but are images of six different cells taken from one section of a new sample block. We now specify this point in the manuscript. We uploaded the original segmentation and raw statistical results for the six images corresponding to Supplementary Fig. 2g to the figshare repository (<https://doi.org/10.6084/m9.figshare.19898404.v3>) for reference.

4. *Another concern that remain unanswered was the question about how the readers are supposed to benefit from DeepContact. Unlike written in the rebuttal letter, I could not find the link to the software in the revised manuscript. Furthermore, the lack of instructions for the use of the software hinders the testing and using of the software by others. Considering that this manuscript is submitted to the Tools section, it lacks very essential parts such as user-friendly GUI and instructions and tutorials for the use of the software.*

We thank the reviewer for the suggestion. As mentioned above, we provide two options by which users can apply the software. The first choice is for users who can employ our models and algorithms directly by programming, which would require basics in computing ([Code available at GitHub: https://github.com/LX-doctorAII/DeepContact](https://github.com/LX-doctorAII/DeepContact)). The other is to incorporate DeepContact into Amira, a popular commercial software for EM image analysis (Tutorial available at [GitHub: https://github.com/LX-doctorAII/DeepContact/blob/main/DeepContact_Tutorial.pdf](https://github.com/LX-doctorAII/DeepContact/blob/main/DeepContact_Tutorial.pdf)), providing a user-friendly GUI for DeepContact. We have added the information in the *Materials and Methods* section of the manuscript.

5. *All images analyzed in the manuscript are taken with in SEM using special sensitive circular backscatter detector, and it is quite clear that boosted membrane contrast is needed for this type of imaging. High membrane contrast is essential for automated segmentation too. Moreover, it remains unclear whether similar workflow could be applied to thin section TEM images, which is the technique that is much more widely accessible for basic EM users. Authors should be more careful in making statements about the suitability of the method for standard contrast samples, unless they show it.*

We appreciate that the reviewer pointed this out. Our initial intention was to enhance the membrane contrast of the images by SEM imaging of ROTO-stained samples for precise segmentation of organelle boundaries by DeepContact. After the initial models were well established, batches of TEM images in different cell lines were also tested by DeepContact, such as ROTO-stained U2OS cells and Cos7 cells and TM4 cells. Incorporation of TEM images with additional training were effective, as represented by Supplementary Fig. 5, for which EM samples with EPG-3/VMP1-, VAPA/B-, and ATL- depletion were kindly provided by other researchers and related image data acquired by TEM. TEM images of liver samples with both ROTO preparation and typical single- osmium tetraoxide-fixation were initially tested by DeepContact after one round of brief training, and promising visualization results were obtained (Fig. 1 a, b), indicating a potential of DeepContact for analyzing standard contrast samples.

Fig. 1 DeepContact segmentation of Mito, ER, and MCS in liver TEM images. a, ROTO prepared liver sample. b, Single-osmium tetroxide-fixation prepared liver sample. Bar, 2 μ m.

6. *Finally as a minor comment, I would like to add that "Sapphire disk-assisted ROTO sample preparation procedure" is confusing way of saying that cultured cells were chemically fixed and embedded as adherent monolayers. I still find no good reason for using sapphires instead of glass coverslips. In my experience, embedding as adherent monolayers is nowadays more common way of processing cultured cells than preparing cell pellets. Comparison between the two sample preparation is unnecessary.*

Considering both Reviewers 2 and 3 have questioned the necessity for introducing "Sapphire disk-assisted ROTO sample preparation procedures" and the necessity for the comparison of adherence monolayer preparation with pallet preparations, we removed the related content from the manuscript and rearranged the Results section and the panels of the figures accordingly.

June 27, 2022

RE: JCB Manuscript #202106190RR

Prof. Li Xiao
Institute Of Computing Technology
No.6 Zhongguancun Road
Haidian District
Beijing, Beijing 100090
China

Dear Prof. Xiao,

Thank you for submitting your revised manuscript titled "DeepContact: High throughput quantification of membrane contact site on electron microscopy imaging." We would be happy to publish your paper in JCB pending final revisions necessary to address one minor comment from Reviewer #3 as well as to meet our formatting guidelines (see details below).

A. MANUSCRIPT ORGANIZATION AND FORMATTING:

- 1) Text limits: Character count for Tools is < 40,000, not including spaces. Count includes title page, abstract, introduction, results, discussion, and acknowledgments. Count does not include materials and methods, figure legends, references, tables, or supplemental legends.
- 2) Figures limits: Tools may have up to 10 main text figures.
- 3) Figure formatting: Scale bars must be present on all microscopy images, including inset magnifications. Molecular weight or nucleic acid size markers must be included on all gel electrophoresis. Please add scale bars to Figures 1, 3a, S1b-e, and S3a-c.
- 4) Statistical analysis: Error bars on graphic representations of numerical data must be clearly described in the figure legend. The number of independent data points (n) represented in a graph must be indicated in the legend. Statistical methods should be explained in full in the materials and methods. For figures presenting pooled data the statistical measure should be defined in the figure legends. Please also be sure to indicate the statistical tests used in each of your experiments (both in the figure legend itself and in a separate methods section) as well as the parameters of the test (for example, if you ran a t-test, please indicate if it was one- or two-sided, etc.). Also, if you used parametric tests, please indicate if the data distribution was tested for normality (and if so, how). If not, you must state something to the effect that "Data distribution was assumed to be normal but this was not formally tested."
- 5) Materials and methods: Should be comprehensive and not simply reference a previous publication for details on how an experiment was performed. Please provide full descriptions (at least in brief) in the text for readers who may not have access to referenced manuscripts. The text should not refer to methods "...as previously described."
- 6) For all cell lines, vectors, constructs/cDNAs, etc. - all genetic material: please include database / vendor ID (e.g., Addgene, ATCC, etc.) or if unavailable, please briefly describe their basic genetic features, even if described in other published work or gifted to you by other investigators (and provide references where appropriate). You must also indicate in the methods the source, species, and catalog numbers/vendor identifiers (where appropriate) for all of your antibodies, including secondary. If antibodies are not commercial please add a reference citation if possible.
- 7) Microscope image acquisition: The following information must be provided about the acquisition and processing of images:
 - a. Make and model of microscope
 - b. Type, magnification, and numerical aperture of the objective lenses
 - c. Temperature
 - d. Imaging medium
 - e. Fluorochromes
 - f. Camera make and model
 - g. Acquisition software
 - h. Any software used for image processing subsequent to data acquisition. Please include details and types of operations

involved (e.g., type of deconvolution, 3D reconstitutions, surface or volume rendering, gamma adjustments, etc.).

8) References: There is no limit to the number of references cited in a manuscript. References should be cited parenthetically in the text by author and year of publication. Abbreviate the names of journals according to PubMed.

9) Supplemental materials: There are strict limits on the allowable amount of supplemental data. Tools may have up to 5 supplemental figures and 10 videos. You currently exceed this limit but, in this case, we will be able to give you some extra space but please try to consolidate the supplemental figures. For example, Figures S4,5,6,7 could be combined into one or at most two figures. Please also note that tables, like figures, should be provided as individual, editable files. A summary of all supplemental material should appear at the end of the Materials and methods section. Please include one brief sentence per item.

10) eTOC summary: A ~40-50 word summary that describes the context and significance of the findings for a general readership should be included on the title page. The statement should be written in the present tense and refer to the work in the third person. It should begin with "First author name(s) et al..." to match our preferred style.

11) Conflict of interest statement: JCB requires inclusion of a statement in the acknowledgements regarding competing financial interests. If no competing financial interests exist, please include the following statement: "The authors declare no competing financial interests." If competing interests are declared, please follow your statement of these competing interests with the following statement: "The authors declare no further competing financial interests."

12) A separate author contribution section is required following the Acknowledgments in all research manuscripts. All authors should be mentioned and designated by their first and middle initials and full surnames. We encourage use of the CRediT nomenclature (<https://casrai.org/credit/>).

13) ORCID IDs: ORCID IDs are unique identifiers allowing researchers to create a record of their various scholarly contributions in a single place. At resubmission of your final files, please consider providing an ORCID ID for as many contributing authors as possible.

B. FINAL FILES:

Thank you for this interesting contribution, we look forward to publishing your paper in Journal of Cell Biology.

Sincerely,

Jodi Nunnari, PhD
Editor-in-Chief
The Journal of Cell Biology

Dan Simon, PhD
Scientific Editor
The Journal of Cell Biology

Reviewer #3 (Comments to the Authors (Required)):

Authors have sufficiently address the questions and comments that were made, and thus I can support publication of their work in JCB.

As a minor point, I would not refer expensive commercial software Amira as "a regularly accessible setup", especially as there are free open access software available for image segmentation and analysis, e.g. Fiji. Authors could consider rephrasing the text on line 90.